# NAD$^+$/NADH redox alterations reconfigure metabolism and rejuvenate senescent human mesenchymal stem cells in vitro

Xuegang Yuan[1,2✉], Yijun Liu[1,4], Brent M. Bijonowski[1,5], Ang-Chen Tsai[1,6], Qin Fu[1,7], Timothy M. Logan[3], Teng Ma[1] & Yan Li[1✉]

Human mesenchymal stem cells (hMSCs) promote endogenous tissue regeneration and have become a promising candidate for cell therapy. However, in vitro culture expansion of hMSCs induces a rapid decline of stem cell properties through replicative senescence. Here, we characterize metabolic profiles of hMSCs during expansion. We show that alterations of cellular nicotinamide adenine dinucleotide (NAD + /NADH) redox balance and activity of the Sirtuin (Sirt) family enzymes regulate cellular senescence of hMSCs. Treatment with NAD + precursor nicotinamide increases the intracellular NAD + level and re-balances the NAD + /NADH ratio, with enhanced Sirt-1 activity in hMSCs at high passage, partially restores mitochondrial fitness and rejuvenates senescent hMSCs. By contrast, human fibroblasts exhibit limited senescence as their cellular NAD + /NADH balance is comparatively stable during expansion. These results indicate a potential metabolic and redox connection to replicative senescence in adult stem cells and identify NAD + as a metabolic regulator that distinguishes stem cells from mature cells. This study also suggests potential strategies to maintain cellular homeostasis of hMSCs in clinical applications.

[1] Department of Chemical and Biomedical Engineering, FAMU-FSU College of Engineering, Florida State University, Tallahassee, FL, USA. [2] Center for Interdisciplinary MR, National High Magnetic Field Laboratory, Tallahassee, FL, USA. [3] Department of Chemistry and Biochemistry, Florida State University, Tallahassee, FL, USA. [4] Amgen Inc., Thousand Oaks, CA, USA. [5] University of Münster, Münster, Germany. [6] University of Florida, Gainesville, FL, USA. [7] Cornell University, Ithaca, NY, USA. ✉email: xy13b@my.fsu.edu; yli4@fsu.edu

n the past decades, human mesenchymal stem/stromal cells (hMSCs) have become attractive candidates for cell therapy, as they exhibit multi-lineage differentiation, paracrine effects, and immunomodulation[1,2]. For clinical applications, preparation of hMSCs with translational and therapeutic standards represents a major effort in cell therapy[1,3,4]. However, inconsistency of clinical results of hMSCs has been observed, possibly due to donor age/morbidity, isolation methods, and extensive in vitro culture expansion[5]. For example, a negative phase III trial of random donor MSCs in steroid-resistant graft-versus-host disease was reported[5,6]. hMSCs from aged or disease donors exhibit reduced stemness and altered therapeutic efficacy including impaired paracrine effects and immunomodulatory function[7–9]. In addition, studies have shown a culture-induced decline of stem cell properties in hMSCs under prolonged culture[1,10]. Extensive passaging of hMSCs reduces the number of colony-forming unit-fibroblasts (CFU-F) and proliferation rate[10–12], which is the hallmark of replicative senescence[13], corresponding with a gradual loss of therapeutic potency in preclinical and clinical studies[1]. However, the mechanism(s) that underpins the culture-induced senescence in hMSC has not been well established.

Recent studies have demonstrated that hMSCs exhibit metabolic plasticity both in vivo and in vitro, which contributes to cellular properties and aging[14,15]. hMSCs exhibit heterogeneity not only at phenotypic level, but also at primary metabolic state determined by their tissue origin[15,16]. Upon isolation of hMSCs from in vivo niche for expansion, in vitro nutrient-enriched environment supports rapid cell proliferation, which requires energy and anabolic macromolecules for daughter cell replication. In this process, catabolic and anabolic pathways are interconnected and together play essential roles in providing energetic sources as well as metabolites to maintain cellular homeostasis[17]. Under this context, hMSCs exhibit metabolic plasticity and can alter their metabolic profile towards efficient oxidative phosphorylation (OXPHOS), which is drastically different from quiescent glycolysis in their in vivo niche[18]. Beyond energetic support, metabolic circuits engage master genetic programs, and intermediate metabolites also mediate cell signaling and regulatory pathways[19]. Thus, metabolic plasticity allows hMSCs to match divergent demands for stem cell properties including self-renewal and differentiation[15,16,19]. Our previous studies indicated specific metabolic reconfigurations of hMSCs in response to certain in vitro preconditioning conditions, which further enhance stem cell functions. For instance, hMSCs under colonial density or three-dimensional aggregation culture reconfigure the energetic metabolism towards glycolysis, and thus improved hMSC stemness[16,20]. However, metabolic alterations associated with replicative senescence in hMSCs during in vitro expansion have not been well investigated.

As hMSCs utilize both glycolysis and OXPHOS as energy source to support their proliferation and regenerative functions, intermediate metabolites could regulate specific signaling pathways[15,16]. In particular, NAD$^+$/NADH redox cycle, which is integral to energy production via glycolysis, tricarboxylic acid (TCA) cycle, and OXPHOS[21], is also a co-substrate of Sirtuin enzymes that regulate cellular homeostasis and lifespan of organism, connecting central energy metabolism to cellular aging and longevity[21–23]. Sirtuins (e.g., Sirt-1) utilize NAD$^+$ to catalyze the deacetylation of histones in target proteins involved in aging process, including P53, poly(ADP-ribose) polymerases (PARPs), the forkhead box O family (FOXOs), nuclear factor-kB (NF-kB), peroxisome proliferator-activated receptor-γ coactivator (PGC)-1α, and Ku70[24,25]. Indeed, NAD$^+$ is the rate-limiting substrate in these Sirtuin pathways[26]. Change to the intracellular NAD$^+$ level has been shown to influence cellular metabolism and Sirtuins associated with aging[27,28]. Other signaling pathways in non-redox

reactions also consume NAD$^+$ to activate downstream functions, such as ADP-ribosyltransferases (e.g., PARPs) and cyclic ADP-ribose synthases (e.g., CD38, CD157), which appear to be the major pathways to reduce the NAD$^+$ level[21,29]. An increasing body of evidence has associated decreasing intracellular NAD$^+$ level and Sirt-1 activity with metabolic diseases and stem cell aging in vivo[30]. Strategies targeting the activity of key enzymes involved in NAD metabolism, such as nicotinamide phosphoribosyltransferase (NAMPT), CD38, and CD73, demonstrate promising results in extending rodent healthspan or lifespan[21,30,31]. Yet, the NAD$^+$/NADH redox balance along with culture expansion, and the relations of NAD$^+$/NADH -Sirtuins and replicative senescence in hMSCs remain to be understood.

Considering the evidence for hMSC metabolic plasticity and the role of NAD$^+$/NADH in metabolism, this study tested the hypothesis that in vitro culture expansion induces replicative senescence and metabolic alterations in hMSCs which correlate with the loss of NAD$^+$ homeostasis and result in the reduction of Sirt-1 signaling activity. In addition, repletion of NAD$^+$ in senescent hMSCs recovers mitochondrial fitness and glycolytic phenotype. Moreover, fully differentiated cell lines such as human dermal fibroblasts (hFBs) do not exhibit culture-induced senescence and changes in NAD$^+$ metabolism during culture expansion, as hFBs and hMSCs are phenotypically similar[32]. Together, our study reveals a novel metabolic and biochemical indicator which can be used for hMSC quality control and rejuvenation in biomanufacturing.

## Results

**hMSCs become senescent and exhibit functional decline during in vitro expansion.** Since hMSCs are density-sensitive, consecutive passaging of hMSCs is necessary to maintain them at the optimal density range (1000–6000 cells/cm$^2$). Our culture protocols for hMSC expansion followed the most widely applied culture strategies (i.e., α-MEM plus 10% fetal bovine serum, 80–90 confluence for harvest etc., details in Methods section). In this study, hMSCs were found to exhibit significant morphological changes and progressive alteration from small, spindle-shaped cells at early passage (e.g., passage 5, referred as P5) to enlarged, flat-shaped cells at late passage (e.g., passage 12, referred as P12) (Fig. 1a). The enlarged cell size of hMSCs indicates increased cellular senescence, which was also characterized by significantly increased SA-β-gal activity (i.e., ~15% for P5 cells vs. 65% for P12 cells) in late passage of hMSCs (Fig. 1b, c). In addition, increased DNA damage was observed using the Comet assay during culture expansion, indicated by increased tail/body length in hMSCs at P12 compared to cells at P5 (Fig. 1d). hMSCs at late passage also exhibited increased population doubling (PD) time (~6 days of P12 vs. 2.8 days of P5) (Fig. 1e), reduced capacity for self-renewal (~10 CFU-F colonies at P12 vs. 96 colonies at P5; Fig. 1f), downregulation of stem cell genes *Oct4* and *Sox2* (but not *Nanog*) compared to early passage cells (P5) (Fig. 1g). The increased levels of mRNA expression for *p53, p15,* and *p21* (Fig. 1h) and accumulation of cells in G0/G1 phase of the cell cycle (Fig. 1i) in P12 hMSCs, indicating a potential cell cycle arrest in hMSCs with replicative senescence. Moreover, downregulation of autophagic genes, including *TFEB, BECN1,* and *LAMP1* were observed in hMSCs at P12 (Fig. 1j), corresponding to the significant decrease of autophagic flux following culture expansion (Fig. 1k). Loss of autophagy indicated the breakdown of cellular homeostasis of hMSCs during in vitro culture expansion. For immunomodulation of hMSCs, mRNA expression of genes (*NF-kB* and *COX2*) involved in chronic inflammation was found to be upregulated in P12 hMSCs compared to P5 cells, while the immunosuppressive ability via IDO pathways in P12 hMSCs decreased (2.76-fold vs. 4.41-fold) under interferon-γ priming (Supplementary Fig. S1). The levels of potent

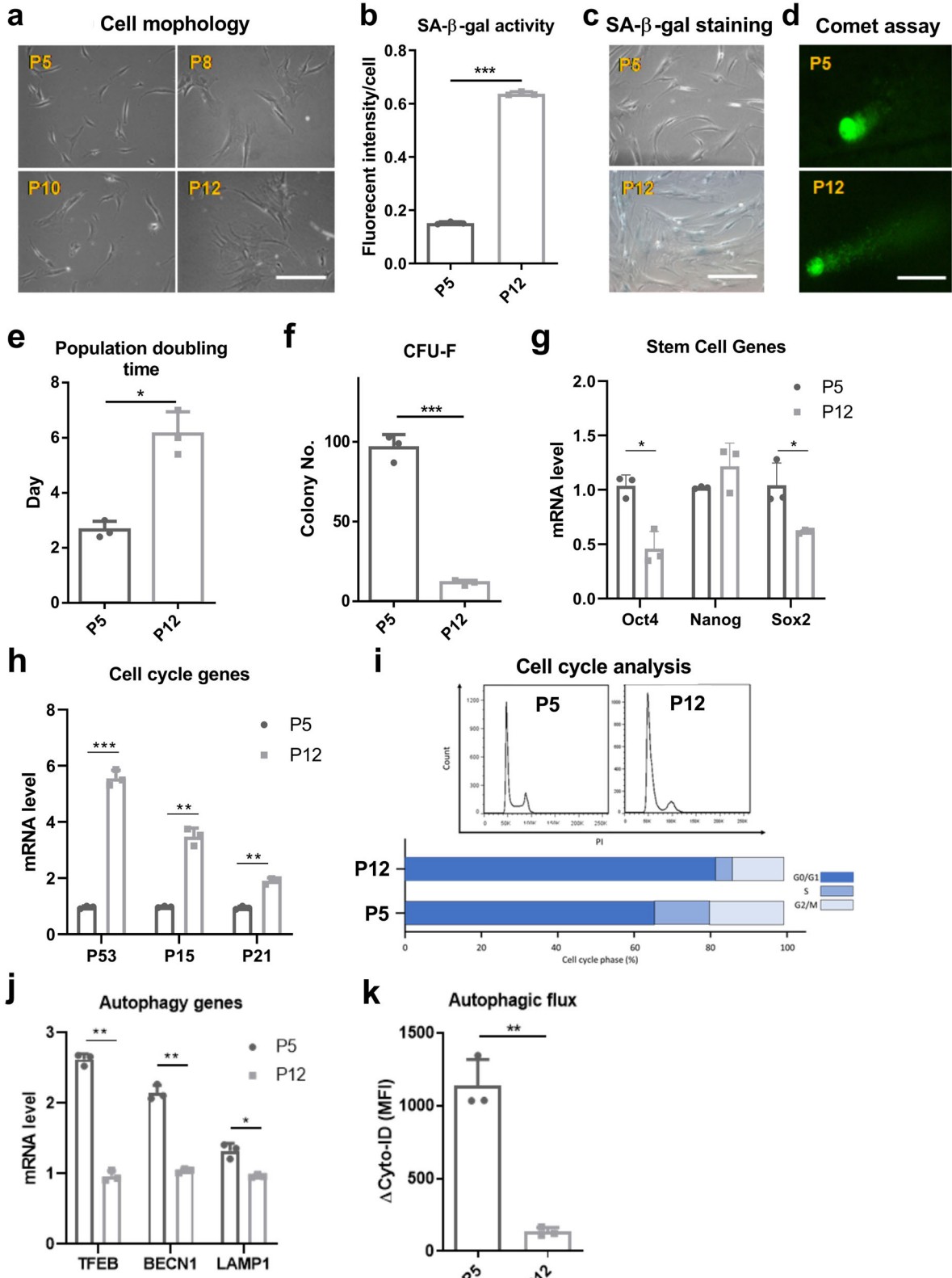

**Fig. 1 In vitro culture expansion of human mesenchymal stem cells (hMSCs) results in cellular senescence and breakdown of cellular homeostasis.**
**a** Alteration of hMSC morphology during culture expansion. **b** SA-β-gal activity and **c** SA-β-gal staining in hMSCs at early passage and late passage. **d** Comet assay demonstrates DNA damage of hMSCs during culture expansion. **e** Population doubling (PD) time increased for long-term cultured hMSCs. **f** Colony-forming unit-fibroblast (CFU-F) ability decreased during culture expansion of hMSCs. **g** mRNA levels of stem cell genes and **h** mRNA levels of cell cycle genes in hMSCs at late passage compared to cells at early passage. **i** Cell cycle analysis of hMSCs via flow cytometry. **j** Autophagic gene expression of hMSCs during culture expansion. **k** Basal autophagic flux was reduced in hMSCs at late passage compared to early passage. Late passage of hMSCs: passage 12 (P12), early passage of hMSCs: passage 5 (P5). Biological replicates (n): n = 3 for the tests. Scale bar: 100 μm. *p < 0.05; **p < 0.01; ***p < 0.001.

anti-inflammatory cytokines HGF and IL-10 decreased while the pro-inflammatory and anti-angiogenic CXCL10 cytokine level significantly increased in late passage of hMSCs. The cytokine level of IL-6, TNF-α, and IL-1β was comparable for P5 and P12 cells (Supplementary Fig. S2). Collectively, these results indicate that hMSCs after prolonged culture expansion enter replicative senescence, which disrupts cellular homeosasis and further reduces hMSC function.

**Culture expansion induces mitochondrial dysfunction in hMSCs.** Consistent with the substantial changes in cell morphology, mitochondrial morphology was also significantly altered during culture expansion: from small-size fragmented morphology in P5 hMSCs towards fused, elongated shape in P12 cells (Fig. 2a). This morphological change was found to correspond to the increase of mitochondrial mass in late passage hMSCs (P12) (Fig. 2b). However, tetramethylrhodamine (TMRM) staining for mitochondrial transmembrane potential (MMP) was found to decrease during culture expansion of hMSCs (Fig. 2c), indicating a loss of membrane integrity and impaired electron transfer ability. The depolarization of mitochondrial membrane was potentially associated with the accumulation of both mitochondrial and total cellular levels of reactive oxygen species (ROS) (Fig. 2d, e) and decreased electron transport chain complex I (ETC-I) activity in hMSCs of late passages (Fig. 2f).

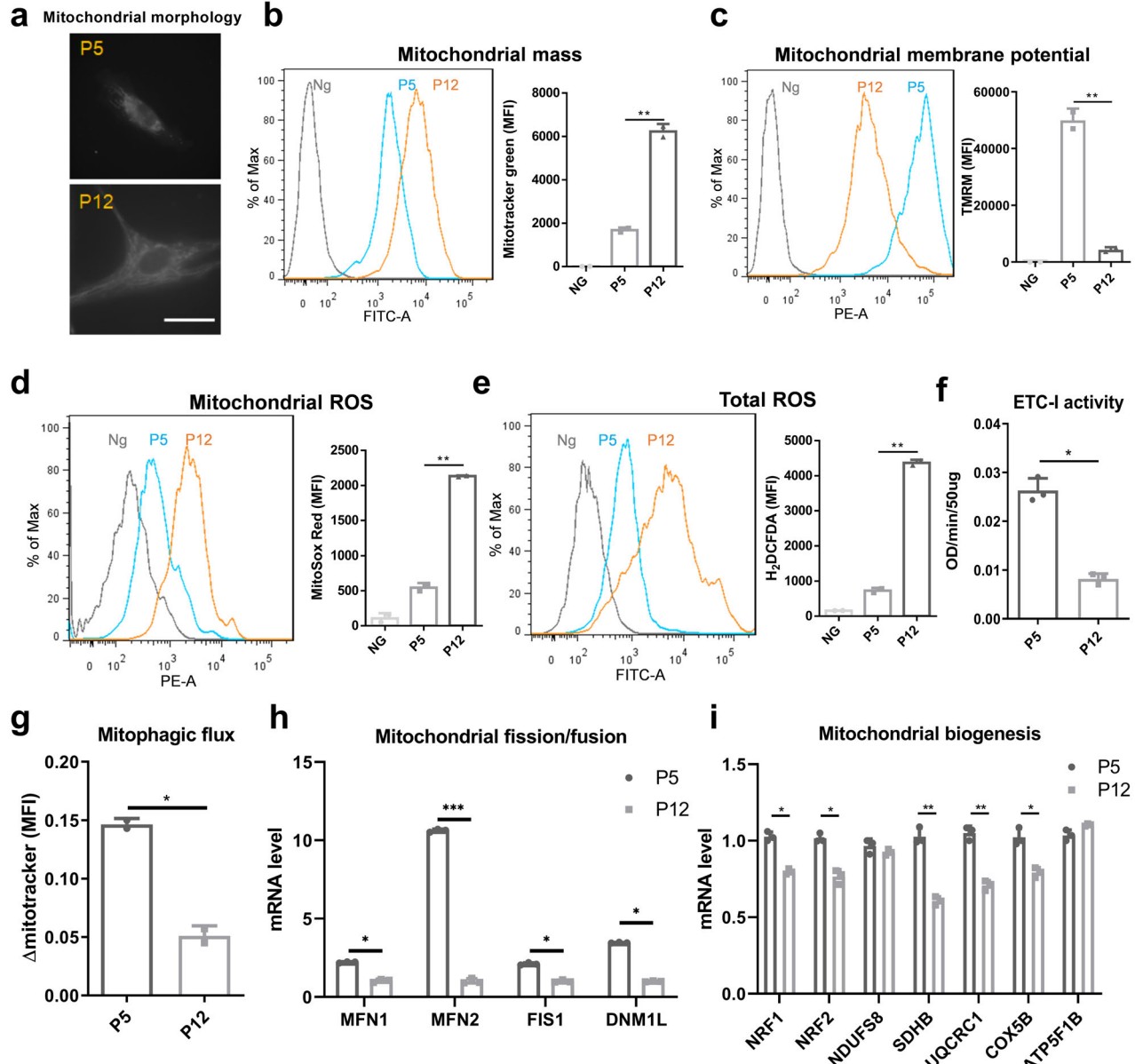

**Fig. 2 Culture expansion of human mesenchymal stem cells (hMSCs) induces mitochondrial dysfunction. a** Mitochondrial morphology was altered during culture expansion of hMSCs. **b** Increased mitochondrial mass and **c** loss of mitochondrial membrane intensity were observed in hMSCs at late passage, both determined by flow cytometry. **d** Mitochondrial reactive oxygen species (ROS) and **e** total ROS were also increased during in vitro culture expansion of hMSCs. **f** Electron transport chain complex I (ETC-I) activity was reduced in late passage of hMSCs compared to cells at early passage. **g** Culture expansion induced loss of hMSC mitophagy, as well as **h** mitochondrial fusion and fission dynamics determined by mRNA levels. **i** Genes involved in mitochondrial biogenesis were decreased during culture expansion of hMSCs. Scale bar: 50 µm. Ng negative control. Biological replicates (*n*): *n* = 3. *p < 0.05; **p < 0.01; ***p < 0.001.

Corresponding to the loss of autophagy, late passage hMSCs exhibited the decreased mitophagic flux compared to early passage cells (Fig. 2g). Expression of genes (e.g., *MFN1*, *MFN2*, *FIS1*, and *DNM1L*) involved in mitochondrial fusion/fission dynamics was downregulated in late passage cells (Fig. 2h), as well as genes involved in mitochondrial biogenesis (e.g., *NRF1*, *NRF2*, *SDHB*, *UQCRC1*, and *COX5B*) (Fig. 2i). Together, these data indicate that replicative senescence induced by long-term culture expansion has substantial impacts on mitochondrial function and biogenesis in hMSCs.

**Culture expansion induces metabolic reconfiguration in hMSCs.** Since mitochondrial fitness is associated with energy metabolism and metabolic plasticity of hMSCs during the adaption of culture environment, the metabolic state of hMSCs at early and late passage was characterized. Glycolytic ATP production was found to be significantly reduced in hMSCs at P12 though slight increase of total ATP was also observed (Fig. 3a). Multiple genes involved in glycolysis (i.e., *PDK1*, *HK2*, *PKM2*, *LDHA*, and *G6PD*) and the pentose phosphate pathway (*6PGD*) were also downregulated in P12 cells (Fig. 3b). By comparison, expression of

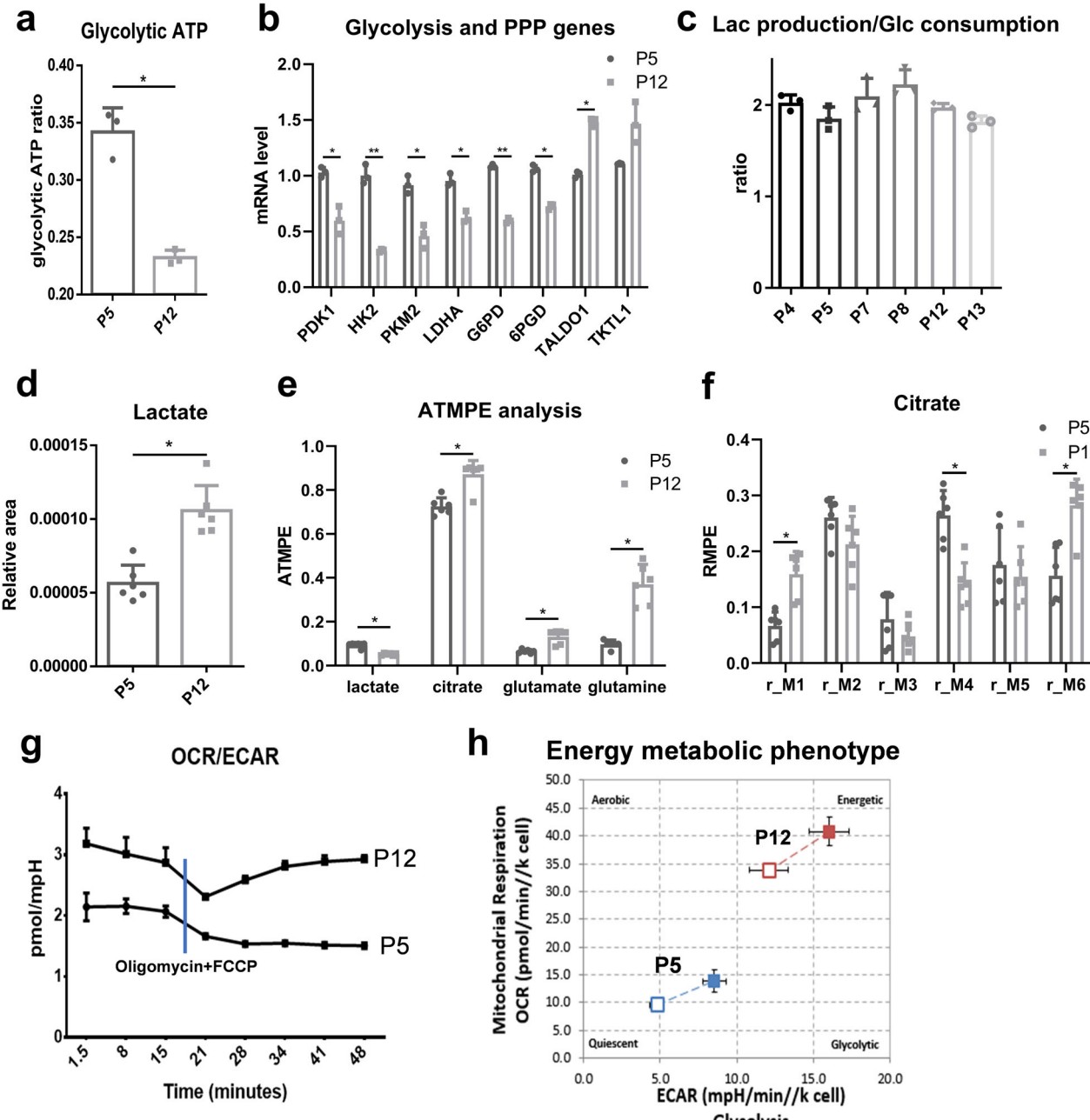

**Fig. 3 Culture expansion induces human mesenchymal stem cell (hMSC) metabolic reconfiguration.** **a** Glycolytic ATP ratio was decreased during culture expansion of hMSCs. **b** mRNA level of genes in glycolysis and pentose phosphate pathways (PPP) of hMSCs during culture expansion. **c** Lactate (Lac) production/glucose (Glc) consumption ratio of hMSCs during extensive expansion. Gas chromatography–mass spectrometry (GC–MS) analysis of hMSCs during culture expansion. **d** Internal normalized peak area of lactate. **e** Absolute total molar percent enrichment (ATMPE) levels of $^{13}$C-glucose atoms in metabolites involved in glycolysis and OXPHOS. **f** The relative molar percent enrichment (RMPE) levels of citrate. **g** OCR/ECAR ratio changes with time of culture in hMSCs at early and late passage. **h** hMSC energy metabolic phenotype during culture expansion based on OCR and ECAR, open square: basal condition; closed square: stressed condition. Biological replicates (*n*): *n* = 6 for GC–MS metabolite study, *n* = 3 for rest of the tests. *$p < 0.05$; **$p < 0.01$.

the trans-aldolase and trans-ketolase enzymes of the pentose phosphate pathway (*TALDO1*) was increased in P12 hMSCs. Interestingly, the lactate production/glucose consumption ratio was relatively stable (around 2.0) across the expansion of hMSCs from P4 to P13 (Fig. 3c). However, gas chromatography–mass spectrometry (GC–MS) analysis of hMSCs at different passages indicated that the internal level of lactate increased in the P12 cells (Fig. 3d). $^{13}$C-glucose tracer experiments showed that the amount of lactate metabolized from glucose decreased, while the amount of citrate increased in P12 hMSCs, suggesting an increased coupling of glycolysis and TCA cycle metabolism as the labeled carbon of glucose has higher enrichment in metabolites from TCA cycle (Fig. 3e). Furthermore, the relative isotopic enhancement of individual carbons in citrate (called isotopomers) did not show significant changes for M2 and M3 between P5 and P12 cells, while the M1, M4, and M6 isotopomers were significantly altered, which may indicate other carbon source (i.e., glutamate and glutamine) instead of glycolysis may also participate in the citrate metabolism (Fig. 3e, f). Whole cell metabolism was also monitored in real time using a Seahorse flux analyzer. It was found that both extracellular acidification rate (ECAR) and oxygen consumption rate (OCR) values increased in P12 cells compared to P5 cells (Supplementary Fig. S3). However, P12 hMSCs exhibited more OXPHOS metabolic profiles after stressed by oligomycin and FCCP (Carbonyl cyanide-4 (trifluoromethoxy) phenylhydrazone), indicating a metabolic shift from glycolysis towards OXPHOS during in vitro culture expansion of hMSCs (Fig. 3g, h).

To understand the global change of proteins involved in metabolism during hMSC culture expansion, proteomic analysis of hMSCs at P4, P8, and P12 was performed to illustrate the global difference in proteome of hMSCs at early passage and late passage with replicative senescence (Supplementary Fig. S4–S13). The experimental results indicate 587 proteins in common (73–79%) across the P4, P8, and P12 hMSCs (Supplementary Fig. S4A). Principal component analysis (PCA) demonstrated a clear separation of P4, P8, and P12 samples as three replicates of each group were clustered (Supplementary Fig. S4B). The 587 proteins were plotted in volcano plots and those with more than 10-fold change and those associated with NAD metabolism (e.g., HADHA, NNMT, HSD17B4, OGDH, and DLD) were marked as yellow (Supplementary Fig. S4C). A total of 80 differentially expressed proteins (DEPs) were selected based on 2-fold cutoff, $p < 0.05$ from the 587 proteins in common. Systematic gene ontology (GO) enrichment analysis illustrated the DEPs to be in three categories: (1) biological process category, primarily enriched in metabolic process, biological regulation, and response to stimulus; (2) cellular component category, primarily enriched in membrane, membrane-enclosed lumen and nucleus; and (3) molecular function category, primarily enriched in protein binding, nucleic acid binding, and nucleotide binding (Supplementary Fig. S5). The DEPs were submitted to the Kyoto Encyclopedia of Genes and Genomes (KEGG) database for intracellular pathway enrichment analysis. A total of nine KEGG pathways were involved, in which the metabolic pathways, glycolysis/gluconeogenesis, cysteine and methionine metabolism, pyruvate metabolism, carbon metabolism, and PI3K-Akt signaling pathway were mainly impacted (Supplementary Table S1 and Fig. S6–S8). These results are consistent with the culture-induced metabolic reconfigurations observed using isotopic tracer methods. Ingenuity Pathway Analysis (IPA) revealed top five canonical pathways that are mostly impacted during culture expansion of hMSCs, including EIF2 signaling, protein ubiquitination, fatty acid beta-oxidation I, 2-ketoglutarate dehydrogenase complex, and superoxide radicals degradation (Supplementary Fig. S9). In particular, fatty acid β-oxidation pathway and 2-ketoglutarate dehydrogenase complex pathway are involved in energy metabolism and NAD$^+$/NADH redox cycle (Supplementary Fig. S10–S13).

**Culture expansion induces NAD$^+$/NADH redox cycle imbalance in hMSCs.** The NAD$^+$/NADH redox cycle plays a crucial role in glycolysis and TCA cycle and also participates in regulation of aging-related signaling pathways and functions[30]. Since culture expansion of hMSCs induces a metabolic reconfiguration of central metabolism, the redox cycle balance may also be affected. Our results indicate that the intracellular NAD$^+$ level progressively declined from P4, P9, to P12 cells, while the NADH level increased during in vitro culture expansion (Fig. 4a). Since the NADH levels increased more than the NAD levels decreased, ratio of NAD$^+$/NADH is lower in P12 hMSCs compared to P4 cells (Fig. 4b). This culture-induced decline of NAD$^+$ level and imbalance of NAD$^+$/NADH were further confirmed in multiple passages and additional donors of hMSCs (Supplementary Fig. S14). It was also found that the protein level and gene expression of Sirt-1 and Sirt-3, the key NAD$^+$-dependent enzymes that control cell signaling pathways and function, were decreased in hMSCs at P12 (Fig. 4c, d, e, f). Sirt-1 regulates mitochondrial biogenesis via PGC-1α and TFAM, which were found to be downregulated in hMSCs at P12 compared to cells at P5. In addition, the gene expression of *PARP1, FOXO1*, and *FOXO3*, which are involved in oxidative stress and regulated by Sirt-1 and Sirt-3, was increased in cells of P12 compared to cells at P5 (Fig. 4g, h). Western blot results confirm the decrease of Sirt-1, Sirt-3, as well as PGC-1 from P5 to P12 at the protein level (Fig. 4i). Together, these results indicate that culture expansion leads to the progressive decline of intracellular NAD$^+$ level and the increase of NADH level, which together change the redox cycle balance in high passage of hMSCs. The reduced expression of NAD$^+$-dependent Sirtuin enzymes was unable to regulate mitochondrial fitness, DNA repair, and other aging-associated pathways during hMSC culture expansion.

**NAD$^+$ biogenesis and metabolism are altered during in vitro culture expansion of hMSCs.** To further investigate the changes of NAD$^+$ and NADH levels, several proteins involved in major pathway of NAD$^+$ biosynthesis and consumption in hMSCs were measured. NAMPT, the rate-limiting enzyme in the NAD salvage pathway that is responsible for maintaining cellular NAD$^+$ levels, was found to significantly increase in hMSCs of P12 compared to cells of P5-6, both at the mRNA level and at the protein level (Fig. 5a, b). hMSCs at P12 also exhibited higher levels of CD38 and CD73, the enzymes responsible for NAD$^+$ consumption, compared to P5 cells (Fig. 5c, d). Western blot analysis confirmed the increase of NAMPT, CD38, and CD73 in senescent hMSCs (Fig. 5e, f). These results demonstrate that culture expansion alters the expression of enzymes responsible for NAD metabolism, which may contribute to the changes in the intracellular NAD$^+$ and NADH levels as well as their ratio.

**Re-balancing NAD$^+$/NADH redox cycle restores mitochondrial fitness and cellular homeostasis in hMSCs with replicative senescence.** Due to the observed relationship between NAD$^+$ metabolism and aging-associated functions in hMSCs (as schematized in Supplementary Fig. S15), re-balancing NAD$^+$/NADH redox cycle may restore mitochondrial fitness and cellular homeostasis, and rejuvenate hMSCs with replicative senescence. Repletion of intracellular NAD$^+$ was achieved by adding the NAD$^+$ precursor, nicotinamide (NAM), to the hMSC culture media as our results demonstrate that there is no NAM in the media (Supplementary Fig. S16). The intracellular NAD$^+$ level and the ratio of NAD$^+$/NADH significantly increased after 96 h

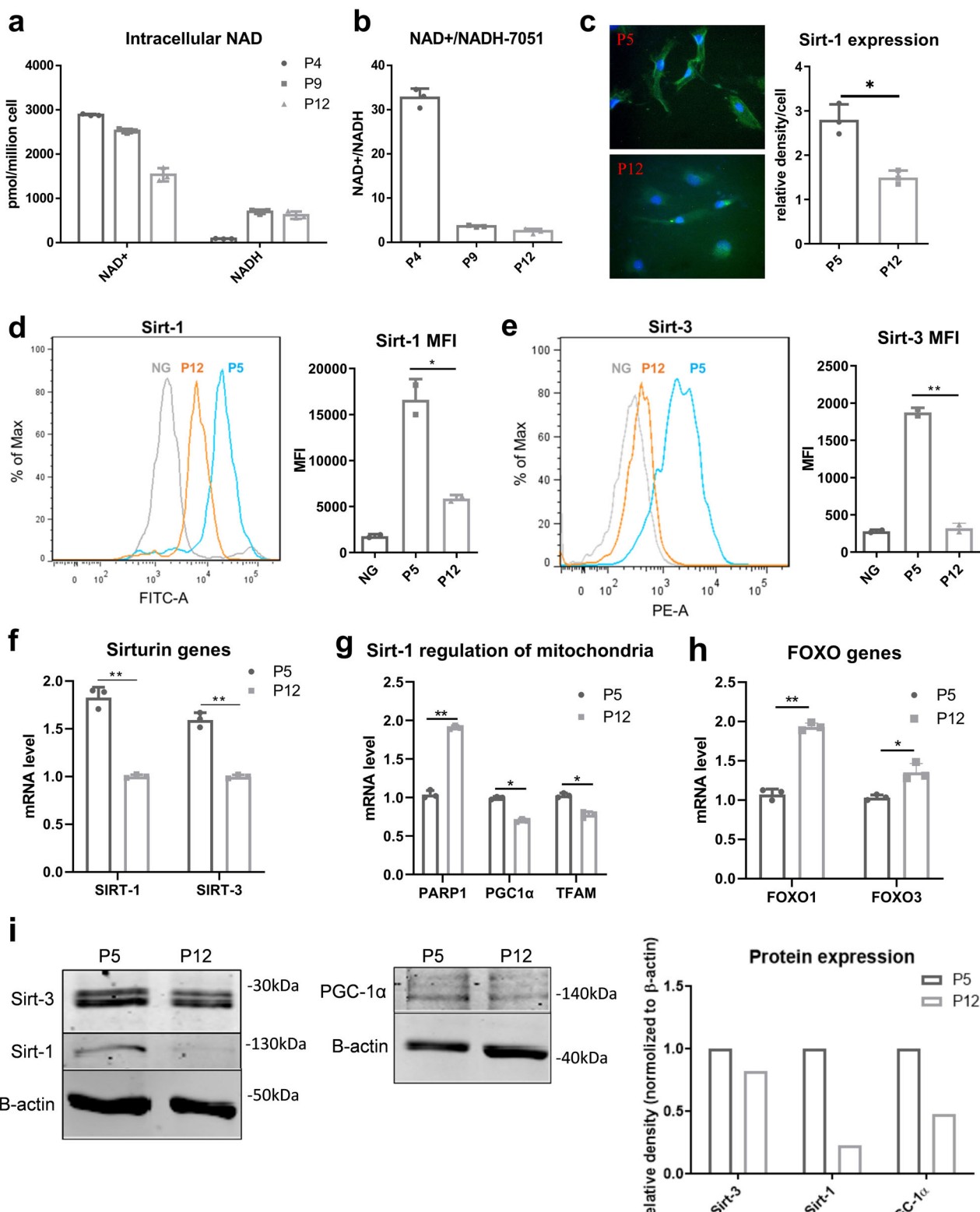

**Fig. 4 NAD$^+$/NADH-Sirtuin (Sirt) axis imbalance induced by in vitro culture expansion of human mesenchymal stem cells (hMSCs). a** Intracellular NAD$^+$ and NADH levels were altered and **b** NAD$^+$/NADH ratio decreased during culture expansion of hMSCs. **c** Immunocytochemistry of Sirt-1 expression in culture-expanded hMSCs. **d** Sirt-1 and **e** Sirt-3 protein levels characterized by flow cytometry. MFI mean fluorescence intensity. **f** mRNA levels of *Sirt-1* and *Sirt-3* in hMSCs. **g** Genes in DNA repair and mitochondria (*PARP1, PGC1*, and *TFAM*) regulated by Sirt-1 in hMSCs at different passages. **h** FOXO pathways (*FOXO1* and *FOXO3*) regulated by Sirt-1 and Sirt-3, determined by RT-PCR. **i** Sirt-1, Sirt-3, and PGC-1α protein levels during culture expansion of hMSCs determined by Western blot. Biological replicates (*n*): *n* = 3. *$p < 0.05$; **$p < 0.01$.

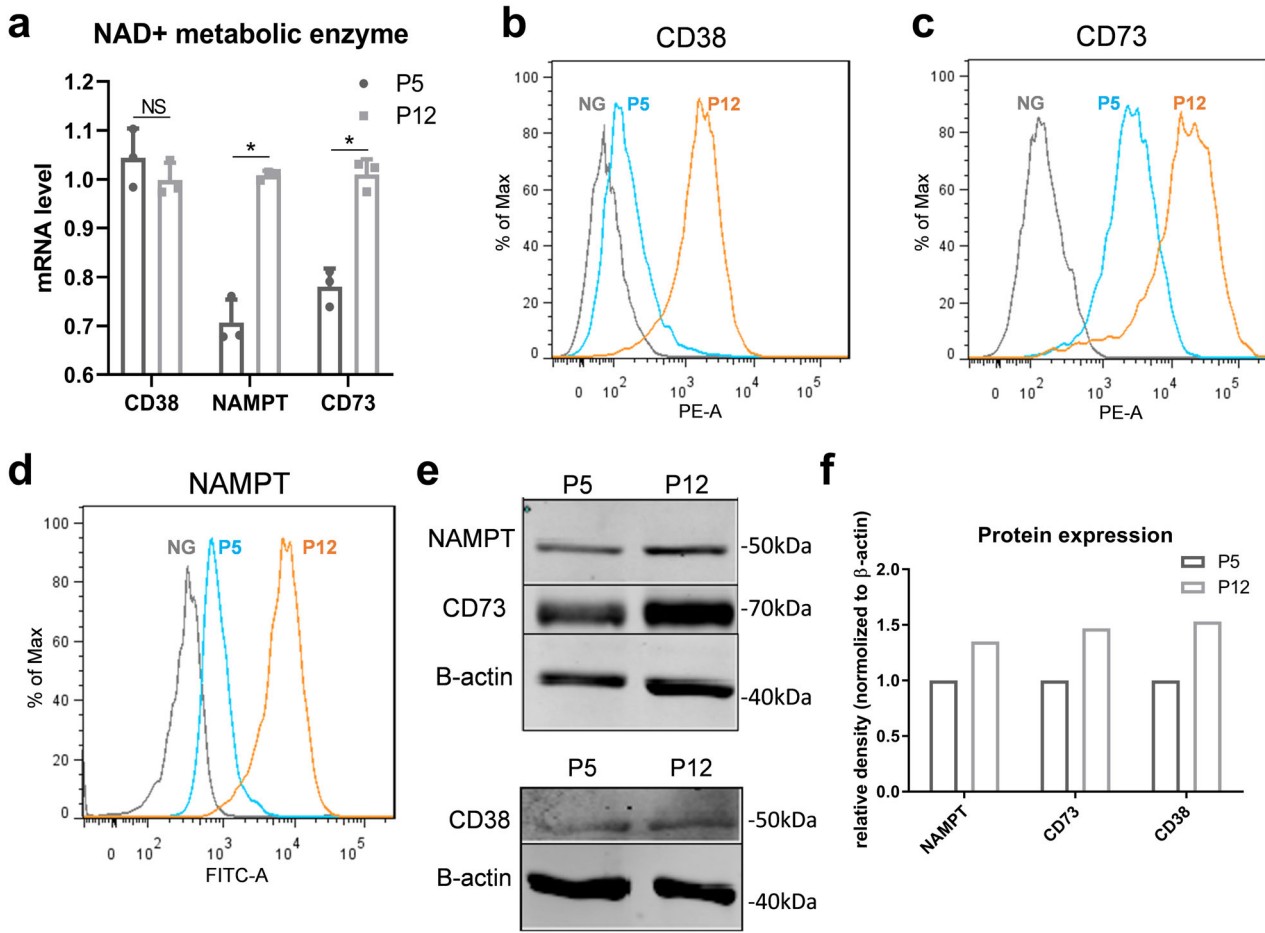

**Fig. 5 In vitro culture expansion of human mesenchymal stem cells (hMSCs) alters NAD$^+$ biogenesis and metabolism. a** mRNA levels of NAD$^+$ metabolic enzymes *NAMPT, CD38,* and *CD73* in hMSCs at early and late passage. **b** NAMPT, **c** CD38, and **d** CD73 protein expressions were all increased in late passage of hMSCs determined by flow cytometry. NG negative control. **e, f** Western blot confirmed the increase of NAMPT, CD38, and CD73 in late passage of hMSCs. Biological replicates (*n*): *n* = 3. *$p < 0.05$. NS not statistically significant.

of NAM treatment (Fig. 6a, b). Sirt-1 and Sirt-3 expression was also increased (Fig. 6c). More importantly, senescence was reduced as indicated by reduced SA-β-gal activity in P12 hMSCs (Fig. 6d). Correspondingly, colony-forming ability was recovered after short-term NAM treatment (from 3 to 11 colonies) (Fig. 6e). After adding NAM, the increase of cell population in the S phase of cell cycle was observed (Supplementary Fig. S17). Consequently, no significant change of population doubling time was announced (data not shown). In addition, the presence of NAM increased glycolytic ATP ratio in P12 cells (Fig. 6f), potentially indicating the increased activity of glycolysis in central metabolism in hMSCs after NAM treatment. Consistent with this, P12 hMSCs with NAM treatment was found to exhibit increased basal autophagy level (Fig. 6g), as well as the reduction of mitochondrial mass (Fig. 6h), re-polarized mitochondrial membrane (increased TMRM staining in Fig. 6i), and increased ETC-I activity (Fig. 6j). The mitophagy was also improved (Fig. 6k) and the total ROS level decreased (Fig. 6l) with NAM treatment in P12 hMSCs. Together, these results indicate that replenishing intracellular NAD$^+$ level and maintaining the NAD$^+$/NADH redox balance in senescent hMSCs increase Sirtuin activity and thus restore mitochondrial fitness and autophagy/mitophagy to maintain cellular homeostasis. These improvements lead to the partial recovery from replicative senescence in hMSCs at high passage, as well as the improved mitochondrial fitness that facilitates central energy metabolism.

**NAD$^+$/NADH redox cycle and mitochondrial fitness are relatively stable during replicative expansion of human dermal fibroblasts.** Since hMSCs exhibit significant changes induced by replicative expansion, similar analysis was then performed for human dermal fibroblasts (hFBs), which were chosen as a representative type of mature adult cells. In contrast to hMSCs, extensive culture expansion (up to passage 15) of hFBs showed no significant difference in population doubling time (Fig. 7a) and SA-β-gal activity (Fig. 7b), indicating that hFBs did not enter cell cycle arrest and become senescent during replicative expansion. Cellular homeostasis was also maintained because no change in autophagic flux was observed (Fig. 7c). hFBs at P15 had similar levels of mitochondrial activity and fitness compared to P4 cells, such as mitochondrial mass (Fig. 7d), transmembrane potentials (Fig. 7e), and mitophagy (Fig. 7f). More interestingly, intracellular NAD$^+$ level and NAD$^+$/NADH ratio was relatively stable during culture expansion up to 15 passages (Fig. 7g). No significant changes were found in Sirt-1 and Sirt-3 expression during the expansion of hFBs as well (Fig. 7h, i). Finally, the expression levels of NAMPT, CD38, and CD73 were comparable throughout the expansion process (Fig. 7j), indicating that NAD$^+$ biosynthesis and metabolism was well maintained in hFBs during replicative expansion. Together, these data indicate that hFBs do not exhibit replicative senescence and are able to preserve cellular homeostasis in artificial culture environment.

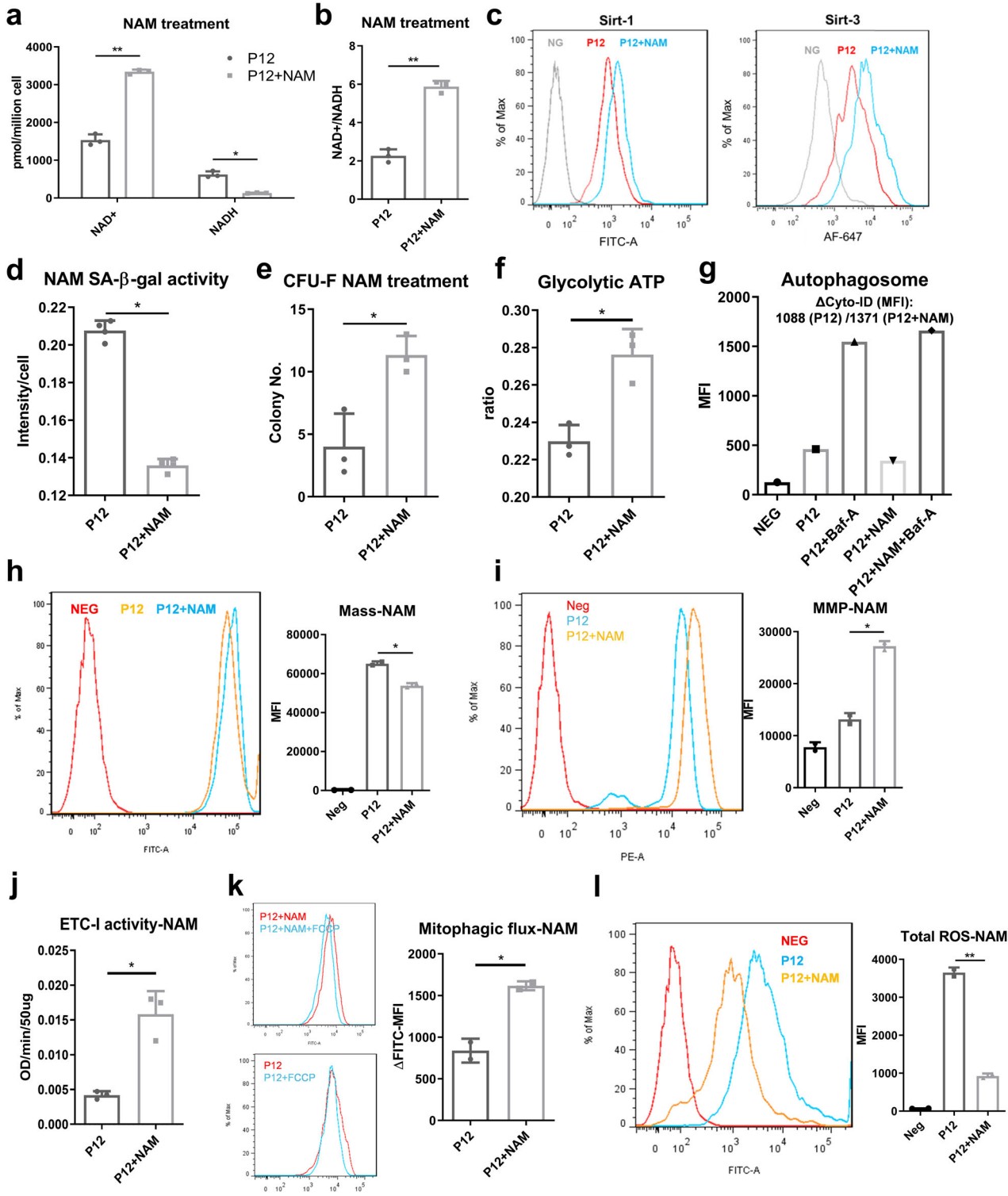

**Fig. 6 Repletion of NAD$^+$ via NAM restores mitochondrial function and preserves stem cell function in long-term cultured human mesenchymal stem cells (hMSCs).** Senescent hMSCs at late passages were treated with NAM for 96 h. **a** NAD level was increased and NADH level was decreased. **b** NAD$^+$/NADH ratio was increased. **c** Sirt-1 and Sirt-3 expressions were both increased. **d** SA-β-gal activity was decreased. **e** Colony-forming ability (CFU-F) and **f** glycolytic ATP ratio were also increased. **g** Basal autophagy was restored in senescent hMSCs after NAM treatment. MFI mean fluorescence intensity. Mitochondrial fitness was restored. **h** Mitochondrial mass was decreased. **i** Mitochondrial transmembrane potential (MMP) and **j** electron transport chain complex I (ETC-I) activity was increased. **k** Mitophagy ability was restored. FCCP mitochondrial uncoupler carbonilcyanide p-triflouromethoxyphenylhydrazone. **l** Total reactive oxygen species (ROS) level was reduced as determined by flow cytometry. Biological replicates (*n*): *n* = 3. \**p* < 0.05; \*\**p* < 0.01.

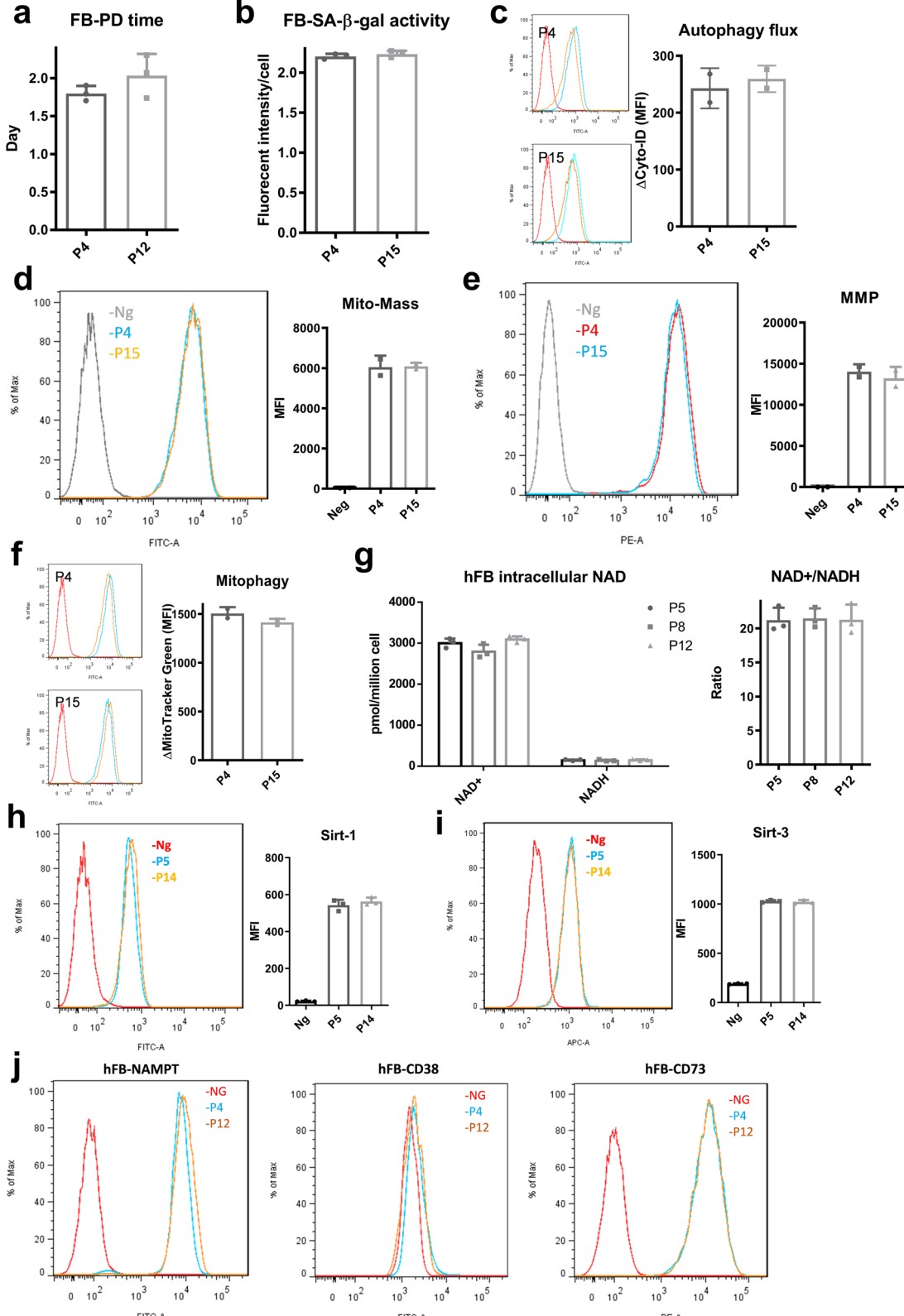

**Fig. 7 Human dermal fibroblasts (hFBs) under replicative expansion exhibit limited cellular senescence, mitochondrial dysfunction, and NAD$^+$ decline.**
**a** Population doubling time of hFBs at early and late passage. **b** SA-β-gal activity for culture-expanded hFBs. **c** No significant difference for autophagic flux in P4 and P15 hFBs. Red line: negative control. Orange line and blue line represents basal autophagy and with autophagy inhibitor Bafliomycin-A (Baf-A), respectively. **d** Mitochondrial mass and **e** mitochondrial transmembrane potential (MMP) showed no difference for P4 and P15 hFBs. **f** Mitophagy also showed no significant difference in hFBs during culture expansion. Red line: negative control. Orange line and blue line represents untreated and with mitochondrial uncoupler (FCCP) treatment, respectively. **g** Intracellular NAD$^+$ and NADH levels, as well as NAD$^+$/NADH ratios are relatively stable during culture expansion of hFBs. **h** Sirt-1 and **i** Sirt-3 protein expression determined by flow cytometry. **j** The expressions of NAD$^+$ metabolic enzymes NAMPT, CD38, and CD73 were all comparable for hFBs at different passages as determined by flow cytometry. Biological replicates (*n*): *n* = 3.

## Discussion

Beyond multi-lineage differentiation, hMSCs exhibit paracrine and immunomodulatory abilities that facilitate endogenous tissue regeneration, thus are recognized as a potential therapeutic candidate. For clinical purpose, in vitro large-scale expansion of hMSCs is a necessary step to meet the requirement for cell number and dosages while maintaining genetic stability and therapeutic efficacy[1]. However, progressive loss of stem cell properties and genetic alterations during in vitro culture of hMSCs have been widely reported[1,10]. This phenomenon is termed as replicative senescence and is the major barrier for hMSCs to be an "off-the-shelf" therapeutic product. This study provides full characterizations of hMSCs during in vitro culture expansion and a novel mechanism underlines replicative senescence, proposing a potential rejuvenation strategy. Particularly, our study reveals the regulatory role of metabolism and intermediate metabolites in hMSC aging and introduces NAD$^+$/ NADH redox balance to connect mitochondrial fitness with replicative senescence in hMSCs.

In vitro culture conditions have been shown to significantly impact cell properties. For hMSCs, a similar Hayflick limit was observed in our studies and by other groups, with altered morphology and arrested cell proliferation following extensive culture[33–37]. In fact, ultra-structure study of cellular organelles revealed endoplasmic reticulum and matrix vesicles were also dysregulated after replicative expansion[38]. Some reports describe a lower ability for differentiation and decreased multipotency of senescent hMSCs. For example, some studies showed increased osteogenesis and decreased adipogenesis, while others showed well-preserved adipogenic potential with diminished osteogenic differentiation[33,34,39], as observed in our study (Supplementary Fig. S18). This may be due to the different medium compositions but still revealed the disruption of multipotency of hMSCs after extensive expansion. In our study, cell cycle arrest in senescent hMSCs was announced by the gradual increase of p53, p21, and p15 gene expression. Though DNA damage was considered as the trigger of p53/p21 signaling pathway to initiate DNA repair, independent activation of p53, p21, and p16 via autophagy and metabolic regulation through AMP-activated protein kinase (AMPK) was also proposed[40–42]. Loss of autophagy has been attributed to functional decline of aged stem cells: for example, aged muscle stem cells and hematopoietic stem cells showed the impaired autophagy along with loss of their regenerative potential, both in vitro and in vivo[43,44]. Restoring autophagy via rapamycin and spermidine treatment partially restored stem cell functions in vivo[43,44]. An interesting phenomenon is that the heterogeneity of aged stem cells also leads to differential autophagic activity[44], potentially explaining the reduced autophagy in hMSCs during culture expansion as heterogeneity increases[45]. For the first time, our results demonstrate a close link between gradual loss of basal autophagy (and mitophagy), a hallmark of cellular homeostasis[46,47], and the replicative senescence in hMSCs.

Our previous studies have extensively demonstrated the metabolic plasticity of hMSCs under artificial culture conditions[15,16,48,49]. Upon removal from the in vivo niche, hMSCs start to adapt to the in vitro environment by utilizing both glycolysis and OXPHOS for ATP production. Cellular homeostasis that contributes to pluripotency and clonal phenotype is well maintained by the low level of glycolytic metabolism for stem cells[15,16]. As metabolism shifts from glycolysis towards primarily OXPHOS, a breakdown of cellular homeostasis is expected due to the accumulation of ROS and damaged organelles. Thus, the metabolic state could also act as a hallmark of replicative senescence during hMSC expansion. As shown in this study, both OCR and ECAR were increased during culture expansion of hMSCs, contributing to the slightly increased ATP production. However,

the ratio of ATP generated from glycolysis gradually decreased, indicating that hMSCs switch their metabolism toward OXPHOS and other pathways to efficiently produce more ATP in order to support extensive replication and maintain stem cell properties[45]. This process may exhaust mitochondrial functions and generate ROS and damaged organelles as autophagy is impaired in senescent hMSCs[50]. Proteomics and metabolomics analysis in current study also reveal the metabolic dysregulations in hMSCs following culture expansion, such as the upregulated fatty acid β-oxidation in late passage hMSCs. This observation may explain the slight increase of ATP production during culture expansion of hMSCs, with gradually decreased percentage of glycolytic ATP. Clearly, hMSCs with extensive culture exhibit metabolic imbalance, which is a hallmark of loss of homeostasis. Moreover, genomics analysis of hMSCs with replicative senescence has demonstrated that genes involved in cell differentiation and apoptosis are upregulated in senescent cells, whereas genes involved in mitosis and proliferation are downregulated[51]. Global omics analysis provides the evidence that changes in replicative senescence may be a general property across hMSC lines regardless of tissue source and culture conditions, but more investigations of the universality of the hMSC replicative senescence in response to extensive culture are still needed[52,53].

NAD$^+$ has been reported to be a regulatory intermediate metabolite associated with aging in yeast and rodents, but few studies focus on its role in in vitro culture of human cells and in vitro senescence[21,54]. Our results demonstrate a progressive decline of intracellular NAD$^+$ level with extensive culture expansion of hMSCs. As a redox cofactor, NAD plays a central role in energy metabolism and also acts as a substrate for enzymes involved in cellular signaling pathways, such as PARPs and Sirtuins[55]. Recently, studies have connected cellular NAD$^+$ level to aging-dependent cell functional decline in multiple organs[56,57]. Replenishment of intracellular NAD$^+$ level was reported to significantly extend the lifespan of yeast, flies, worms, and mice[58–62]. It was reported that NAD$^+$ supplement rejuvenates senescent muscle stem cells and neural stem cells in aged mice via regulation of mitochondrial fitness and the unfolded protein response to improve metabolic activity[30]. Interestingly, our study did not observe significant change in AMPK activity, an energy gauge for sensing energetic alteration. Thus, culture-induced replicative senescence of hMSCs may differ from in vivo stem cell aging in the context of energy production. For hMSCs, a metabolic shift towards OXPHOS contributes to the changes in the NAD$^+$/NADH redox balance as NAD$^+$ is rapidly converted to NADH by TCA cycle (by isocitrate dehydrogenase, α-ketoglutarate dehydrogenase, and malate dehydrogenase). Generally, NADH would be oxidized back to NAD$^+$ through the electron transport chain[63]. However, our results show that senescent hMSCs with damaged mitochondria exhibit depolarized mitochondrial membrane and impaired ETC-I functions, which could compromise the ability to metabolize NADH and maintain NAD$^+$/NADH redox balance. This observation thus indicates that senescent hMSCs may shift towards more reducing (or less oxidized) NAD$^+$/NADH cycle. Notably, decreased ETC activity generally leads to decreased OCR, which is opposite in our observations. This may be due to other cellular events also regulating basal OCR, such as ATP turnover, proton leak, and non-mitochondrial oxygen consumption (ROS formation)[64]. In addition, loss of autophagy/mitophagy further creates a feedback loop that facilitates the redox imbalance in our study. Accumulated DNA damage and continuous activation of PARP during culture expansion could also facilitate the NAD$^+$ depletion in senescence hMSCs[65–68].

Two major mechanisms in aging-related NAD$^+$ decline have been proposed: (1) decline of NAD$^+$ biosynthesis; and (2)

increase of NAD[+] consumption by enzymes competing with each other for the cellular NAD[+] pool or chronological aging-induced enzymatic dysfunctions[25]. To better understand the factors contributing to the metabolism of NAD[+] in senescent hMSCs, several key enzymes involved in NAD[+] biosynthesis and consumption were investigated in this study. CD38 and CD73 were highly upregulated, indicating the enhanced consumption of NAD[+] in hMSCs with replicative senescence. Similar to the way that CD38 is upregulated in immune cells under inflammatory environment, hMSC senescence can be attributed to chorionic inflammation following in vitro culture[69,70]. To our surprise, NAMPT, a rate-limiting intermediate enzyme in the major salvage pathway for NAD[+] biogenesis, is also highly upregulated in hMSCs with replicative senescence. This finding is seemingly contradictory to the decline of NAD[+] level but can be explained by the absence of NAD[+] biosynthetic substrates (or NAD precursors) such as NAM. Consistent with this explanation, NAM supplement improves intracellular NAD[+] level and mitochondrial fitness in senescent hMSCs, even with short-term treatment in our study. Based on these results, the NAD[+]/Sirt-1 axis dysfunction could be a potential checkpoint for the loss of stemness and breakdown of homeostasis in adult stem cells, and can be restored by supplying NAD[+] precursors for biosynthesis.

This study further examined whether human dermal fibroblasts exhibit similar changes during culture expansion. Surprisingly, within a similar number of population doublings, hFBs exhibit a relatively consistent cell growth and β-gal activity, indicating that the cellular senescence did not increase following the expansion of hFBs. Moreover, NAD[+]/NADH redox balance as well as Sirt-1/Sirt-3 activity was well maintained in hFBs at late passage. Mitochondrial function and autophagy/mitophagy were also comparable throughout culture expansion in hFBs. These results demonstrate that hMSCs and hFBs have different sensitivities to artificial culture environment under in vitro expansion. Generally, fibroblasts were considered to share similar phenotypic characteristics with MSCs[32,71,72], including lineage-specific differentiation and colony-forming ability, though these properties are highly donor- and tissue source-dependent[71,73]. Studies have revealed that fibroblasts can be cultured for 60–80 population doublings before entering replicative senescence[74], making them much more replicative compared to hMSCs. Moreover, hFBs do not exhibit metabolic reconfiguration under the nutrient-enriched culture environment. In fact, switching to anaerobic metabolism mostly occurs in response to serum starvation rather than reducing oxygen level in hFB culture[75]. By comparison, hMSCs are extremely sensitive to their culture environment including nutrients, oxidative stress, mechanical stimuli, or even gravity[76]. hMSCs are able to adapt to different culture environments and stimuli (e.g., hypoxia and cytokine potentiation) to maintain stemness and cellular function. The adaption process, in most cases, is required for engineering hMSCs with enhanced therapeutic potentials[77]. In fact, this sensitivity provides the possibility to engineer hMSCs with culture conditions instead of genetic modification. For instance, hypoxia, 3D aggregation, and cytokine priming can enhance hMSC properties for clinical purposes via metabolic reconfiguration[15,49,78–80]. hFBs, however, may not be engineered by the metabolic preconditioning since they are less sensitive to the artificial culture environment.

hMSCs under in vitro culture expansion exhibit replicative senescence, accompanied with functional decline and loss of homeostasis, which is regulated by a metabolic shift and a change of NAD[+]/NADH redox balance. In addition, mitochondrial damage and loss of autophagy also contribute to the replicative senescence of hMSCs during expansion. Our results demonstrate that repletion of NAD[+] via its precursor NAM re-activates NAD[+]/Sirt-1 axis to improve mitochondrial fitness and further restores cellular homeostasis in hMSCs with replicative senescence. This observation suggests a simple strategy for manipulating culture conditions for biomanufacturing to maintain desired therapeutic quality in hMSC-based therapy.

## Methods

**hMSC and hFB cultures**. Frozen hMSCs from passage 0 to 2 were acquired from Tulane Center for Gene Therapy. The hMSCs were isolated from the bone marrow of multiple healthy donors with age 19–49 years old based on plastic adherence, negative for CD3, CD14, CD31, CD45, and CD117 (all less than 2%) and positive for CD73, CD90, CD105, and CD147 markers (all greater than 95%) and possess tri-lineage differentiation potential upon in vitro induction[48,81]. Informed consent was obtained from all research participants. All experimental procedures and ethical regulations have been reviewed and approved by Office for Human Subjects Protection & Institutional Review Board in Florida State University. hMSCs (1 × 10[6] cells/mL/vial) in freezing media containing α-MEM, 2 mM L-glutamine, 30% fetal bovine serum (FBS), and 5% dimethyl sulfoxide (DMSO) were thawed and cultured following the method described in our prior publications[16,49,80,82]. Briefly, hMSCs were expanded and maintained in complete culture media (CCM) containing α-MEM with 10% FBS (Atlanta Biologicals, Lawrenceville, GA) and 1% Penicillin/Streptomycin (Life Technologies, Carlsbad, CA) in a standard incubator at 37 °C with 5% $CO_2$ and 20% $O_2$. Culture medium was changed every three days. Cells were grown to 70–80% confluence and then harvested by incubation with 0.25% trypsin/ethylenediaminetetraacetic acid (EDTA) (Invitrogen, Grand Island, NY) at 37 °C for 4–7 min. Harvested cells were re-plated at a density of 1500 cells/cm[2] and subcultured up to passage 15. For comparison of cells at different passages, hMSCs from the same source were used.

Primary human dermal fibroblasts (containing mitochondria, PCS-201-012™) were purchased from American Type Culture Collection (ATCC, Manassas, VA) and subcultured in CCM up to passage 15. All reagents were purchased from Sigma Aldrich (St. Louis, MO) unless otherwise noted.

**Cell number, CFU-F, SA-β-Gal activity, comet assay, and glucose/lactate measurements**. Cell number was determined by Quant-iT™ PicoGreen kit (Invitrogen, Grand Island, NY). Briefly, cells were harvested, lysed overnight using proteinase K (VWR, Radnor, PA), and stained with Picogreen to allow quantitation of cellular DNA. Fluorescence signals were measured using a Fluor Count (PerkinElmer, Boston, MA). Population doubling time (mean PD time) was determined through culture in each passage:

$$\text{Mean PD time} = \frac{t}{\log_2 n}$$

where $t$ is culture time and $n$ is the cell number fold increase during culture time $t$.

For CFU-F assay, hMSCs were harvested and re-plated at the density of 15 cells/cm[2] on 60 cm[2] culture dish and cultured for another 14 days in CCM. Cells were then stained with 20% crystal violet solution in methanol for 15 min at room temperature (RT) and gently washed with phosphate-buffered saline (PBS) for three times. The number of individual colonies were counted manually. Cellular senescence was evaluated by SA-β-Gal activity assay kit (Sigma, St. Louis, MO) as described in manufacturer's instructions.

Fresh and spent CCM were collected to determine glucose consumption and lactate production by YSI 2500 Biochemistry Select Analyzer (YSI,Yellow Spring, OH). Cellular DNA damage was measured by comet assay (Cell Biolabs, Inc. San Diego, CA), following manufacturer's instructions.

**Mitochondrial morphology, mass and membrane potential, and ROS level measurements**. For mitochondrial morphology, hMSCs were incubated with 100 nM MitoTracker Red CMXRos (Molecular Probe, Eugene, OR) in CCM at 37 °C for 30 min. After washing with PBS, cells were fixed with 3.7% formaldehyde at 37 °C for 15 min and then imaged with Olympus IX70 microscope.

For mitochondrial mass and MMP measurement, trypsinized hMSCs were washed in warm Hank's Balanced Salt Solution (HBSS). Cell suspension was incubated with MitoTracker green FM or tetramethylrhodamine, methyl ester (Molecular Probe, Eugene, OR) at 37 °C for mass and MMP staining, respectively. Cells were then washed with HBSS and analyzed by flow cytometry (BD Biosciences, San Jose, CA).

For ROS measurement, cell suspension was incubated with 25 μM carboxy-H2DCFDA (Molecular Probe) at 37 °C for 30 min and total ROS was determined using flow cytometry. For mitochondrial ROS measurement, cell suspension was incubated with 5 μM MitoSOX Red (Molecular Probe) at 37 °C for 10 min and analyzed using flow cytometry.

**Immunocytochemistry, cell cycle, autophagy, and mitophagy measurements**. Cells were harvested with 0.25% trypsin-EDTA solution, washed in PBS, and then fixed at 4% paraformaldehyde (PFA) at RT for 15 min. Cells were then permeabilized in 0.2% triton X-100 for 10 min at RT. Non-specific binding sites were blocked with 1% bovine serum albumin, 10% FBS in PBS for 15 min at RT. After washing, cells were incubated with specific primary antibodies for human Sirt-1,

Sirt-3, NAMPT, CD38, and CD73 (Santa Cruz Biotechnology, Dallas, TX) at RT for 2 h, followed by incubation with FITC-conjugated secondary antibody (Molecular Probe). Labeled samples were analyzed by flow cytometry. Antibody information was summarized in Supplementary Table S2. Gating strategy for hMSCs at early and late passage was demonstrated in Supplementary Fig. S19.

For cell cycle analysis, suspended cells were fixed with 70% cold ethanol for 30 min at 4 °C and then washed with PBS. 100 μg/mL RNase A (VWR, Radnor, PA) was added to cell suspension and incubated at 35 °C for 15 min. Then the samples were incubated with 400 μL 50 μg/mL of propidium iodide (VWR) solution at RT in the dark for 1 h. Cell cycle was then determined by flow cytometry.

For autophagy measurement, cell suspension was incubated with 20 μM Cyto-ID Green (Enzo Life Sciences, Farmingdale, NY), a fluorescent dye that selectively labels accumulated autophagic vacuoles, at 37 °C for 30 min, and analyzed by flow cytometry and calculated according to the manufacturer's instructions. For mitophagy measurement, the cells were incubated with mitochondrial uncoupler carbonilcyanide p-triflouromethoxyphenylhydrazone (FCCP, 1 μM) for 20 min and then the mitochondrial mass was tested via flow cytometry. Mitophagic flux in the cells was calculated by the different mitochondrial mass between treated and untreated group.

**Intracellular ATP content, mitochondrial complex I activity, and metabolic phenotype**. hMSCs were centrifuged, re-suspended in deionized water, and heated immediately in boiling water for 15 min. The mixture was centrifuged, and ATP-containing supernatant was collected. Upon measurement, 10 μL of ATP solution was mixed with 100 μL of the luciferin-luciferase reagent (Sigma-Aldrich), and the bioluminescent signal was measured using an Orion Microplate Luminometer (Bad Wildbad, Deutschland). To determine the glycolytic ATP ratio, cells were cultured with or without glycolysis inhibitor 2-Deoxy-D-glucose (2-DG, 5 mM) for 48 h and then the ATP was measured. The ratio of glycolytic ATP was calculated by the delta value of total ATP and 2-DG treated ATP normalized to total ATP. Activity of mitochondrial electron transport complex I activity was determined using the Complex I Enzyme Activity Microplate Assay Kit (Abcam, Cambridge, MA) according to the manufacturer's instructions.

OCR and ECAR were determined using Agilent Seahorse XF Extracellular Flux Analyzer XFp (Seahorse Biosciences, Massachusetts, USA). All tests were performed in accordance with manufacturer's instructions. Briefly, hMSCs were seeded onto Seahorse XFp Cell Culture Miniplate (Seahorse Biosciences) at 10,000 cells per well the day before being analyzed. Cells were equilibrated in a non-CO$_2$ incubator with Seahorse calibrant buffer for 60 min prior to assay. Using the Seahorse XFp Cell Energy Phenotype Test Kit (Seahorse Biosciences), OCR and ECAR under baseline and stressed conditions (oligomycin and FCCP) were measured[83].

**Intracellular NAD$^+$ and NADH quantification**. Intracellular NAD$^+$ and NADH were measured with NAD$^+$/NADH Quantification Colorimetric Kit (BioVision, Milpitas, CA) according to manufacturer's instructions with some modifications. Briefly, approximate 0.8 million cells were collected and directly lysed in 200 μL lysis buffer from the assay kit. The volume of reagents in each step was scaling down by 50% and the results were calculated by the freshly prepared standard curve (NADH standards provided by the assay kit). Final NAD$^+$ and NADH concentrations were then normalized to the total cell number in each group.

**$^{13}$C-glucose labeling and GC–MS analysis of hMSC metabolites**. $^{13}$C-glucose labeling, metabolite extraction, and chemical derivatization were performed as in our prior publications[16,48]. Briefly, glucose-free DMEM medium supplemented with a 2:3 mixture of unlabeled and U-$^{13}$C- labeled glucose (Cambridge Isotopes Laboratories, Andover, MA) at the same concentration as CCM for hMSC expansion (1.0 g/L glucose). P5 and P12 hMSCs were seeded and cultured for 2 days in DMEM with unlabeled medium. The culture medium was then replaced with isotope-enriched medium and cultured for additional 3 days. Cell collection started by washing with PBS, quenching with liquid nitrogen, and addition of a solution of methanol:water (4:1) directly to the culture plate to stop metabolism and lyse the cells on dry ice, followed by addition of the internal standard (norleucine 28 μg/ml solution) and incubation at −80 °C for 10 min. The extracts were then centrifuged at 5000 x $g$ for 5 min at 4 °C, and the supernatants were collected and transferred to a silanized Reacti-Vial (Wheaton) and stored at −80 °C. Prior to derivatization, frozen extracts were dried under vacuum overnight and dissolved in 20 μL pyridine and 20 μL N-methyl-N-(tert-butyldimethylsilyl) trifluoro-acetamide containing 1% tert-butyldimethylchlorosilane (Thermo Scientific, Rockford, IL). The reaction was performed under a stream of argon. Reacti-vials were closed under argon and heated to 75 °C for 60 min and then cooled to room temperature. Injection of derivatized extracts in the GC–MS was completed within 24 h of derivatization.

Derivatized samples (1 μL) were injected in splitless mode at 230 °C in an HP Agilent 6890 series gas chromatograph (GC) coupled with an HP Agilent 5973 mass selective detector and separated on a 30 m DB5 column (J&W Scientific, Folsom, CA). The GC oven temperature was held at 70 °C for 1 min after injection, increased to 120 °C at 15 °C min$^{-1}$ and finally to 325 °C at 10 °C min$^{-1}$. Mass

spectra were collected over m/z 50–650 at a rate of 2 Hz with MS source set at 260 °C. Metabolites were identified by comparison with standards. Peak areas were calculated from the [M-57]+• and [M-159]+• ions for amino acids and [M-57]+• and [M-189]+• ions for carboxylic acids by fitting the elution profile of each isotopomer to a Gaussian, eliminating the baseline, and summing over all isotope peaks for each specific ion[84,85]. The area was then normalized to the peak area of the internal standard norleucine, and divided by the cell count. Mass isotope distribution vectors and isotope incorporation was determined using methods described in detail elsewhere[48].

**Real-time reverse transcriptase-polymerase chain reactions**. Total RNA was isolated using the RNeasy Plus kit (Qiagen) following vendor's instructions. Reverse transcription was carried out using 2 μg of total RNA, anchored oligo-dT primers (Operon), and Superscript III (Invitrogen). Primers for specific target genes were designed using the software Oligo Explorer 1.2 (Genelink) (Supplementary Table S3). β-actin was used as an endogenous control for normalization. RT-PCR reactions were performed on an ABI7500 instrument (Applied Biosystems), using SYBR Green PCR Master Mix. The amplification reactions were performed and the quality and primer specificity were verified. Fold variations in gene expressions were quantified using the comparative Ct method: $2^{-\Delta(CtTreatment -CtControl)}$, which is based on the comparison of the target gene (normalized to β-actin) among different conditions.

**Statistics and reproducibility**. Unless otherwise noted, all experiments were performed at least in triplicate ($n = 3$), and representative data are reported. Experimental results are expressed as means ± standard deviation (SD) of the samples. Statistical comparisons were performed by one-way ANOVA and Tukey's post hoc test for multiple comparisons, and significance was accepted at $p < 0.05$.

## Data availability statement

The datasets generated during and/or analyzed during the current study are available from the corresponding authors on reasonable request. All source data underlying the graphs and charts presented in the main figures are available on Figshare: https://doi.org/10.6084/m9.figshare.13198187.v1. The mass spectrometry proteomics data have been deposited to the ProteomeXchange Consortium via the PRIDE partner repository with the dataset identifier PXD022395.

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

## Acknowledgements

The authors would like to thank Ms. Ruth Didier of FSU Department of Biomedical Sciences for her help in flow cytometry and Dr. Brian Washburn and Kristina Poduch of FSU Department of Biomedical Sciences for performing RT-PCR experiments. The authors would also like to thank Translational Science Laboratory at Florida State University for help in proteomics experiments. This work was supported by National Science Foundation Award (CBET #1743426) and partially supported by the National Institutes of Health (NIH; R01 NS102395). The content is solely the responsibility of the authors and does not necessarily represent the official views of the NIH.

## Author contributions

X.Y., Y.L., A.C.T., T.M., and Y.L. conceived the experiments and wrote the manuscript, X.Y. conducted the majority of the experiments, B.B. performed Western blot analysis, Q.F. performed sample preparation and data analysis for proteomics results, T.L. helped in NMR and LC-MS experiments and reviewed the manuscript, and X.Y., T.M., and Y.L. analyzed results.

## Competing interests

The authors declare no competing interests.
