## [Peer Review File · Communications Biology]

Reviewers' Comments:

Reviewer #1:

Remarks to the Author:

MS# COMMSBIO-20-0427-T

Li et al. NAD⁺/NADH redox

In this manuscript, the authors reported their recent work on the role of NAD⁺/NADH redox cycle and metabolic changes in the regulation of senescence of human mesenchymal stem cells (hMSCs). They used an in vitro cell culture system to demonstrate the metabolic characteristics of hMSCs in lower and higher passages and the regulatory association of metabolic reconfiguration in replicative senescence. The research was well designed and performed, generating a lot of interesting data that support, if not all, the conclusions. However, in the current form, there are several drawbacks that need to be addressed.

1. Cell passages and synchronization. Human bone marrow-derived MSCs were cultured and used in the study of senescence. Cell markers were examined and heterogeneity was identified. Harvested cells were re-plated at a density of 1,500 cells/cm² and sub-cultured up to passage 15. For comparison of cells at different passages, hMSCs from the same source were used. Because the authors were investigating the impact of cell expansion on senescence, the cell cultures should be synchronized to a G1/G0 phase in cell cycle, before conducting any experiments.
2. The authors stated that alterations in the total pool size and redox balance of cellular nicotinamide adenine dinucleotide (NAD⁺) and the lowered activity of the Sirtuin (Sirt) family enzymes were found to be highly associated with cellular senescence and breakdown of cellular homeostasis during culture expansion of hMSCs. The results are interesting. However, it is unclear about the level of oxygen consumption and if the oxygen consumption relate to ATP production in the NAD⁺/NADH redox cycle in the different cell passages.
3. Treatment with NAD⁺ precursor nicotinamide (NAM) was used as a culture supplement to boost the intracellular NAD⁺ level and maintain the NAD⁺/NADH balance. The authors found that the NAM treatment increased Sirt-1 activity, with partially recovered indicators of overall mitochondrial fitness. They believe the treatment rejuvenates hMSCs exhibiting replicative senescence. Cell cycle comparison is needed to demonstrate the recovery or rejuvenation from replicative senescence.
4. Human dermal fibroblasts were purchased from ATCC and cultured as control cells. Do the fibroblasts contain mitochondria? The cells lacking the mitochondria might not be a proper control for this replicative senescence study.
5. The authors claimed they were using "stem cells" but did not show evidence of cell differentiation into different mature cell lineages. It would be nice should the authors provide the evidence of cell maturation in function and morphology.

Minor points:

Throughout the figures, there is an inconsistency: in some experiments, the authors used P4, P12 cells but others P5, P14. The inconsistency is confusing and casting the question of data collection and analysis.

Reviewer #2:

Remarks to the Author:

The authors address a significant problem that hMSCs undergo a limited number of cell divisions in vitro and then enter replicative senescence characterized by growth arrest. The main finding that

alterations to the total pool size and redox balance of cellular nicotinamide adenine dinucleotide (NAD⁺) and the lowered activity of the Sirtuin (Sirt) family enzymes were found to be highly associated with cellular senescence. The paper is remarkable in presenting more than 35 kinds of measurements in the first 5 figures to provide evidence that hMSCs replicative senescence is altered in ways related to redox balance, some of which are not affected in human dermal fibroblasts. In addition, the supplementary figures provide a wealth of proteomics analysis. Together, these findings led them to supplement the culture medium for 3 days with nicotinamide as a precursor to NAD. Importantly, this nicotinamide treatment rejuvenates 12 of these measures of metabolic reconfiguration. A number of issues arise that could improve this important work.

Major points

1. Line 40 of abstract is not supported by their data, NAM treatment "maintained the NAD⁺/NADH balance." See Fig. 4B compared to 6B.
2. Line 237 concludes that "culture expansion leads to the progressive decline of intracellular NAD⁺ level." While true, the greater fold effect appears to be an increase in NADH to change the redox status.
3. line 273 and Fig 6J. How can the nearly 4-fold increase in complex I activity in the ETC by NAM be interpreted as "partially reconfigure hMSC metabolism to the glycolytic phenotype."
4. The title is misleading. NAM did the rejuvenation. The metabolic reconfiguration as a function of passage was well documented by changes at the transcriptional, proteomic and enzymatic levels. Only a limited subset of these studies after NAM treatment were presented. Also, what is the redox cycle? Certain elements of redox reactions were demonstrated, but no clear single cycle.
5. Fig 3G, H show elevated OCR with passage. The small increase in glycolytic ATP with NAM in Fig. 6F seems too small to claim on line 273 that NAM can "partially reconfigure hMSC metabolism to the glycolytic phenotype seen in low passage cells." Where's the OCR measure after NAM?

Minor points

1. In the intro., line 84, the authors state, "Our previous studies indicated specific metabolic reconfigurations in response to certain in vitro preconditioning conditions of hMSCs". A little more explanation would help.
2. line 109, The hypothesis that culture-induced hMSC senescence is correlated with changes in NAD⁺ homeostasis should be more specific as to which direction and kind of homeostasis. E.g. lower pool size or oxidized shift in NAD/NADH ratio.
3. line 122, please summarize "the most widely applied method with standardized culture medium."
4. Images supplied for Fig. 1A, C are poor quality, lack contrast and blue SA-beta-gal stain is barely visible. In 1B, can you give us an indication of what percent of cells are beta-gal positive?
5. Supplementary Table 1 needs a comparison column of non-senescent cells and a reference in line 137 to what CD90 and CD105 are. If this is an n of 1, remove this data. If more than 1, include statistics.
6. Fig. 1 and all figures needs the n stated for each panel. Otherwise, the statistics are adequate. Attention to the large magnitude of some of the differences in the results may increase biological significance.
7. Fig 2A is unintelligible with lines through the image.
8. Fig 2C, flow cytometry shows a decrease in signal for P12, but the TMRM signal increases. The TMRM increase suggests mitochondria are more hyperpolarized. It is not clear whether this number has been normalized for the increase in mitochondria mass.
9. line 156. Since MMP is negative, "decrease" is ambiguous. Use depolarized or hyperpolarized.
10. All labels in Figure 3C are too small to read. Use font like in 3B.
11. There are no methods to describe how NAD and NADH were measured.
12. There are no methods to describe how ETC-I activity was measured.
13. Line 226 should be revised to include "since the NADH levels increased more than the NAD levels

decreased, ... (a significantly lower ratio of NAD/NADH).

14. Line 269 change increase MMP to hyperpolarize or repolarize MMP. Also need to normalize to mitochondrial mass.

15. Discussion line 321-322 states "For the first time, our results demonstrate a close link between gradual loss of basal autophagy and mitophagy, a hallmark of cellular homeostasis 44, 45, and the replicative senescence in hMSCs." A summary of this finding belongs in the abstract as well.

16. Line 357 states "eventually lose the ability to maintain NAD⁺/NADH redox balance." If lost, they would die. Instead, they shift to a more reducing, less oxidized redox balance.

17. Fig 5A part of legend is cropped for passage number.

18. Lines 369-374 discuss the decline in NAD levels despite a rise in NAMPT. Could it be that the rise in NAMPT is insufficient to metabolize the larger rises in Sirt1 and Sirt3?

19. The methods should state the culture gases, probably 5% CO₂ and 20% oxygen (hyperoxia). The discussion should mention that this alone could account for much of the physiologic response. Rather than stating the advantages of hypoxia in the discussion, they might be more accurately termed normoxia.

20. Some comment in the discussion may be needed about why OCR goes up but ETC-I levels go down.

21. Supplementary figure S15 needs the colors explained in a legend or figure caption.

Response to Reviewers' Comments

The authors appreciate the thoughtful and constructive comments by the reviewers. We have thoroughly revised the manuscript and added detailed information/discussions to address the comments expressed by the reviewers. In addition, we performed additional experiments to the best we can. The detailed responses to the comments are summarized in the following section and the revisions in the manuscript and supplementary information are highlighted in red color. The reviewers' comments are reproduced in *italic font*, and our responses are in regular font.

Reviewer #1 (Remarks to the Author): MS# COMMSBIO-20-0427-T Li et al. NAD⁺/NADH redox

In this manuscript, the authors reported their recent work on the role of NAD⁺/NADH redox cycle and metabolic changes in the regulation of senescence of human mesenchymal stem cells (hMSCs). They used an in vitro cell culture system to demonstrate the metabolic characteristics of hMSCs in lower and higher passages and the regulatory association of metabolic reconfiguration in replicative senescence. The research was well designed and performed, generating a lot of interesting data that support, if not all, the conclusions. However, in the current form, there are several drawbacks that need to be addressed.

1. Cell passages and synchronization. Human bone marrow-derived MSCs were cultured and used in the study of senescence. Cell markers were examined and heterogeneity was identified. Harvested cells were re-plated at a density of 1,500 cells/cm² and sub-cultured up to passage 15. For comparison of cells at different passages, hMSCs from the same source were used. Because the authors were investigating the impact of cell expansion on senescence, the cell cultures should be synchronized to a G1/G0 phase in cell cycle, before conducting any experiments.

Response:

We thank the reviewer for the comments. In this study, we try to imply the most practical approaches of *in vitro* culture expansion in laboratory for most of the hMSC studies, which is culturing hMSCs on tissue culture plastics and passaging hMSCs when they reach 80-90% confluency and identifying as “one passage”. At this time frame (80-90% confluency), hMSCs were collected for all the tests. We choose this approach because hMSCs exhibit heterogeneity after isolation from tissue source, meaning that there are several cell populations existing in each harvest. Thus, hMSCs were likely in different cell cycle stages during any culture time point. It is very challenging to determine the cell cycle for each population of hMSCs at different passages. Moreover, clinical application of hMSCs for transplantation also imply similar culture and harvest strategies as we did. Thus, in our study, we choose to test the expanded hMSCs when they proliferate for roughly two population doublings and reach 80-90% confluency.

2. The authors stated that alterations in the total pool size and redox balance of cellular nicotinamide adenine dinucleotide (NAD⁺) and the lowered activity of the Sirtuin (Sirt) family enzymes were found to be highly associated with cellular senescence and breakdown of cellular homeostasis during culture expansion of hMSCs. The results are interesting. However, it is unclear about the level of oxygen consumption and if the oxygen consumption related to ATP production in the NAD⁺/NADH redox cycle in the different cell passages.

Response:

The individual measurement of OCR and ECAR of different passages of hMSCs corresponding to their metabolic phenotype has been added to the supplementary information as **Supplementary Figure S3**. Clearly, the OCR and ECAR were all increased when hMSCs were expanded *in vitro* (as well as total ATP production). However, the ATP increase is almost neglectable which indicated that the low efficient energy metabolism in high passage of hMSCs due to the loss of NAD⁺/NADH redox cycle and the breakdown of metabolic homeostasis in this study. More importantly, when taken the ratio of OCR/ECAR, hMSCs at high passage (P12) was significantly higher than hMSCs at low passage (P5) as shown in Figure 3G. This indicated the metabolic shift towards OXPHOS during culture expansion of hMSCs. The metabolic reconfiguration acts as one key step in the loss of NAD⁺/NADH redox homeostasis. Thus, increased OCR is associated with ATP production and metabolic shift, which also indicate the breakdown of NAD⁺/NADH redox balance by exhausting mitochondrial functions, which negatively impact the cellular homeostasis. More discussion on this observation has been added to the revised manuscript.

3. Treatment with NAD⁺ precursor nicotinamide (NAM) was used as a culture supplement to boost the intracellular NAD⁺ level and maintain the NAD⁺/NADH balance. The authors found that the NAM treatment increased Sirt-1 activity, with partially recovered indicators of overall mitochondrial fitness. They believe the treatment rejuvenates hMSCs exhibiting replicative senescence. Cell cycle comparison is needed to demonstrate the recovery or rejuvenation from replicative senescence.

Response:

We have performed additional experiment to analyze the cell cycle after 96 hr NAM treatment on the late passage of hMSCs. This result has been added as **Supplementary Figure S17**. We observed a slight increase of S phase and decrease in G0/G1 phase in the senescent hMSCs after NAM treatment. However, this improvement may not be significantly announced on the population doubling time. Please also see the response to the editor's point #2.

4. Human dermal fibroblasts were purchased from ATCC and cultured as control cells. Do the fibroblasts contain mitochondria? The cells lacking the mitochondria might not be a proper control for this replicative senescence study.

Response:

For clarification, we purchased the human dermal fibroblast from ATCC. This fibroblast cell line contains mitochondria. As the reviewer pointed out, we included this information in the revised "Methods" section.

5. The authors claimed they were using "stem cells" but did not show evidence of cell differentiation into different mature cell lineages. It would be nice should the authors provide the evidence of cell maturation in function and morphology.

Response:

As the reviewer suggested, we have performed trilineage differentiation on hMSC cell line used in this study. The results have been added as **Supplementary Figure S18**. Also, please refer to

our previous publications for the visualization of trilineage differentiation of our hMSC lines (Rosenberg et al. 2013; Kim et al. 2012; Grayson et al. 2007). Please see our response to the Editor's point #3.

References:

1. Rosenberg et al. Magnetic resonance contrast and biological effects of intracellular superparamagnetic iron oxides on human mesenchymal stem cells with long-term culture and hypoxic exposure. *Cytotherapy*. 2013, 15(3): 307-322.
2. Kim et al. Autocrine fibroblast growth factor 2-mediated interactions between human mesenchymal stem cells and the extracellular matrix under varying oxygen tension. *Journal of cellular biochemistry*. 2012, 114(3): 716-727.
3. Grayson et al. Hypoxia enhances proliferation and tissue formation of human mesenchymal stem cells. *Biochemical and biophysical research communications*. 2007, 358(3): 948-953.

Minor points:

Throughout the figures, there is an inconsistency: in some experiments, the authors used P4, P12 cells but others P5, P14. The inconsistency is confusing and casting the question of data collection and analysis.

Response:

For clarification, most of our experiments compared hMSCs at P4 and P12. However, during actual operation (during the past three years), sometimes the scheduling cannot fit with P4 and P12 cells, we have to compare P5 and P14 cells. As the passage numbers are close enough for low passage (P4-P5) and high passage (P12-P14) cells, we think the results should reflect the effects of ageing. We apologize for this variation, but we think it is more important to record the actual information during the experiments.

Reviewer #2 (Remarks to the Author):

The authors address a significant problem that hMSCs undergo a limited number of cell divisions in vitro and then enter replicative senescence characterized by growth arrest. The main finding that alterations to the total pool size and redox balance of cellular nicotinamide adenine dinucleotide (NAD⁺) and the lowered activity of the Sirtuin (Sirt) family enzymes were found to be highly associated with cellular senescence. The paper is remarkable in presenting more than 35 kinds of measurements in the first 5 figures to provide evidence that hMSCs replicative senescence is altered in ways related to redox balance, some of which are not affected in human dermal fibroblasts. In addition, the supplementary figures provide a wealth of proteomics analysis. Together, these findings led them to supplement the culture medium for 3 days with nicotinamide as a precursor to NAD. Importantly, this nicotinamide treatment rejuvenates 12 of these measures of metabolic reconfiguration. A number of issues arise that could improve this important work.

Major points

1. Line 40 of abstract is not supported by their data, NAM treatment “maintained the NAD⁺/NADH balance.” See Fig. 4B compared to 6B.

Response:

Firstly, we would like to thank the reviewer for all comments and suggestions. As the reviewer suggested, the abstract has been carefully revised in the manuscript, regarding NAM treatment: “Treatment with NAD⁺ precursor nicotinamide (NAM) as a culture supplement increased the intracellular NAD⁺ level and re-balanced the NAD⁺/NADH ratio, with enhanced Sirt-1 activity in hMSCs at high passage, partially recovered overall mitochondrial fitness and rejuvenated hMSCs with replicative senescence.”

2. Line 237 concludes that “culture expansion leads to the progressive decline of intracellular NAD⁺ level.” While true, the greater fold effect appears to be an increase in NADH to change the redox status.

Response:

As the reviewer suggested, this statement has been revised in the manuscript to reflect the changes in both NAD⁺ and NADH for the alteration of redox cycle. The summary sentence of Figure 4 is now “Together, these results indicate that culture expansion leads to the progressive decline of intracellular NAD⁺ level and increased NADH level, which together change the redox cycle in high passage of hMSCs. The reduced expression of NAD⁺-dependent Sirtuin enzymes was unable to regulate mitochondrial fitness, DNA repair and other aging associated pathways during hMSC culture expansion.”

3. line 273 and Fig 6J. How can the nearly 4-fold increase in complex I activity in the ETC by NAM be interpreted as “partially reconfigure hMSC metabolism to the glycolytic phenotype.”

Response:

In Fig 6, the ETC-I activity (Fig. 6J), together with mitochondrial membrane potential (Fig. 6I), represent the partial recovery of mitochondrial fitness, while the glycolytic ATP represent more glycolytic activity for energetic metabolism. The glycolytic ATP in hMSCs with senescence is not boosted in a higher manner as we expected after 96 h treatment of NAM. This is the reason that we conclude that there may only be a partial reconfiguration of glycolysis after the short-term treatment of NAD⁺ precursors. This statement has been revised to be more precisely reflective for the conclusion of our findings. Please also see the response to the editor’s point #4.

4. The title is misleading. NAM did the rejuvenation. The metabolic reconfiguration as a function of passage was well documented by changes at the transcriptional, proteomic and enzymatic levels. Only a limited subset of these studies after NAM treatment were presented. Also, what is the redox cycle? Certain elements of redox reactions were demonstrated, but no clear single cycle.

Response:

As the reviewer suggested, the title has been revised.

We agree that limited experiments have been done for the metabolic profile after the NAM treatment. Since we only tested the short-term treatment of NAM and only observed limited improvement of glycolytic activity, this set of experiments is more like proof-of-concept study to indicate that NAD⁺ repletion could rejuvenate or, maintain the cellular homeostasis of hMSCs

during culture expansion. This provides the potential for using NAD⁺ precursors as culture medium supplement to better preserve hMSC quality for large-scale expansion and we have observed the long-term effect of NAD⁺ precursor supplement in culture medium to delay metabolic alteration and senescence of hMSCs (manuscript in preparation). In current study, we want to demonstrate that the NAD⁺/NADH redox cycle connects stem cell central metabolism, mitochondrial fitness and *in vitro* aging and could act as an indicator of senescence to evaluate hMSC quality in biomanufacturing.

The redox cycle in this study is:

The redox cycle of NAD⁺/NADH is referred to the global cellular NAD⁺ and NADH conversion and transportation that drives energetic metabolism. For instance, glycolysis converts NAD⁺ to NADH (from glucose to pyruvate) and oxidizes NADH back to NAD⁺ via lactate dehydrogenase in cytosol. For carbon flow into the TCA cycle, NAD⁺ is converted to NADH via pyruvate dehydrogenase to generate Acetyl-CoA. NAD⁺ is constantly converted to NADH (by isocitrate dehydrogenase, α -Ketoglutarate dehydrogenase, and malate dehydrogenase) during the whole TCA cycle in mitochondria. Moreover, NADH from cytosol can be transferred to mitochondrial membrane via malate-aspartate shuttle and together with the NADH generated by TCA cycle can be converted to NAD⁺ via NADH/coenzyme Q reductase located at electron transport chain complex-I. Each mole of NADH converted by ETC-I generates 3-mole of net ATPs. This NAD⁺/NADH conversion and transportation is considered as NAD⁺/NADH redox cycle in hMSCs during culture expansion. Notably, NAD⁺ consumption and biosynthesis is considered as NAD⁺ metabolism in this study. This mechanism is actually well studied, and the details are not included in the background introduction of this manuscript due the word limit.

5. Fig 3G, H show elevated OCR with passage. The small increase in glycolytic ATP with NAM in Fig. 6F seems too small to claim on line 273 that NAM can “partially reconfigure hMSC metabolism to the glycolytic phenotype seen in low passage cells.” Where’s the OCR measure after NAM?

Response:

Please also see the response to the editor’s point #5.

This statement has been revised in the manuscript to better summarize our findings. Now the summary sentences of Figure 6 are “Together, these results indicate that replenishing intracellular NAD⁺ level or maintaining the NAD⁺/NADH redox balance in senescent hMSCs improves Sirtuin activity and thus restore mitochondrial fitness and autophagy/mitophagy to maintain cellular homeostasis. These improvements lead to the partial recovery from replicative senescence in high passage of hMSCs, as well as a glycolytic energy phenotype as seen in low passage cells.” We agree that double check the OCR and ECAR after NAM treatment would be interesting to understand the general impact on hMSCs’ metabolism. However, the Seahorse analyzer that was used to measure OCR/ECAR is not available to us anymore. The flux measurements in this manuscript were performed more than one year ago during an instrument demonstration. Now, this instrument is not available anymore, and due to Coronavirus pandemic and restricted budget, it is unlikely that our institution will acquire this instrument. We apologize for this situation.

Minor points

1. In the intro., line 84, the authors state, “Our previous studies indicated specific metabolic reconfigurations in response to certain *in vitro* preconditioning conditions of hMSCs”. A little more explanation would help.

Response:

As the reviewer suggested, the following statements with the details on metabolic reconfiguration of hMSCs under preconditioning culture has been added to the revised manuscript: “For instance, hMSCs under colonial density or three-dimensional aggregation culture reconfigure the energetic metabolism towards glycolysis, and thus improved hMSC stemness.”

2. line 109, *The hypothesis that culture-induced hMSC senescence is correlated with changes in NAD⁺ homeostasis should be more specific as to which direction and kind of homeostasis. E.g. lower pool size or oxidized shift in NAD/NADH ratio.*

Response:

As the reviewer suggested, we re-phrased the hypothesis in the revised manuscript with more specific changes of NAD⁺/NADH as follow: “this study tested the hypothesis that culture-induced senescence of hMSCs is correlated with changes in the loss of NAD⁺ homeostasis, which results in the reduction of Sirt-1 signaling.”

The reason we use “NAD⁺ homeostasis” is that we focused not only on the intracellular level of NAD⁺ and NADH, but the whole metabolism of NAD⁺ and NADH within hMSCs during culture expansion. The biosynthesis, consumption, and conversion of NAD⁺ were studied and discussed in this study to elucidate the loss of NAD⁺ homeostasis during *in vitro* culture expansion of hMSCs, which eventually results in cellular senescence.

3. line 122, *please summarize “the most widely applied method with standardized culture medium.”*

Response:

As the reviewer suggested, we revised this sentence as “Our culture protocols for hMSC expansion followed the most widely applied method (i.e., α -MEM plus 10% fetal bovine serum, details in Methods section).”

Specifically, the culture strategies in this study imply the most common culture conditions used for hMSCs or other adult stem cells: cells were seeded at the density of 1,500 cells/cm² and culture for roughly 2-3 population doublings. Then the hMSCs were passaged for sub-culture. Cells were cultured in a standard incubator at 37°C with 5% CO₂ and 20% O₂. The culture medium used in this study contains α -MEM with 10% FBS and 1% Penicillin/Streptomycin. Medium was changed every 3 days. More details are also provided in the Methods section in the revised manuscript.

4. *Images supplied for Fig. 1A, C are poor quality, lack contrast and blue SA-beta-gal stain is barely visible. In 1B, can you give us an indication of what percent of cells are beta-gal positive?*

Response:

As the reviewer suggested, Figure 1A and 1C have been revised with images with high resolution. We want to clarify that Figure 1A and 1C look normal in word file of the main text. However, the PDF file conversion in the submission system significantly reduced the image quality. If this happens again for the PDF conversion for the revised manuscript submission, we will directly send the original images for the manuscript production.

In addition, the quantification of β -gal activity is shown in Figure 1B. As the reviewer suggested, we provided the quantitative numbers in the results section to indicate the increase of senescence, which reads as “...., which was characterized by significantly increased SA- β -gal activity (i.e., about 15% for P5 cells vs. 65% for P12 cells) in late passage of hMSCs...”

5. Supplementary Table 1 needs a comparison column of non-senescent cells and a reference in line 137 to what CD90 and CD105 are. If this is an n of 1, remove this data. If more than 1, include statistics.

Response:

As the reviewer suggested, we removed this data since only one sample of cell line was tested. The major goal is to demonstrate the expression of the surface makers has shifted slightly but still follow the criteria of hMSCs established by International Society for Cellular Therapy (ISCT). The manuscript has also been revised.

6. Fig. 1 and all figures needs the n stated for each panel. Otherwise, the statistics are adequate. Attention to the large magnitude of some of the differences in the results may increase biological significance.

Response:

As the reviewer suggested, n numbers for the biological replicates are included in each figure legend in the revised manuscript.

7. Fig 2A is unintelligible with lines through the image.

Response:

For clarification, Figure 2A looks normal in word file of the main text. However, the PDF file conversion in the submission system significantly reduced the image quality. As the reviewer suggested, we provide the original Figure 2A with high image quality in the revision. If this happens again for the PDF conversion for the revised manuscript submission, we will directly send the original images for the manuscript production.

8. Fig 2C, flow cytometry shows a decrease in signal for P12, but the TMRM signal increases. The TMRM increase suggests mitochondria are more hyperpolarized. It is not clear whether this number has been normalized for the increase in mitochondria mass.

Response:

We apologize for the confusion. The passage number was mislabeled in the original figure. As the reviewer pointed out, Figure 2C now has been revised. The TMRM or MMP is not normalized to mitochondrial mass.

9. line 156. Since MMP is negative, “decrease” is ambiguous. Use depolarized or hyperpolarized.

Response:

We appreciated the suggestions. The MMP statement has been revised to describe the polarization of the mitochondrial membrane.

10. All labels in Figure 3C are too small to read. Use font like in 3B.

Response:

As the reviewer suggested, Figure 3C now has been revised.

11. There are no methods to describe how NAD and NADH were measured.

Response:

We apologize for not including the methods for measuring NAD⁺ and NADH. The details for the assay kit and measurement were added in the revised Materials and Methods.

12. There are no methods to describe how ETC-I activity was measured.

Response:

For clarification, the ETC-I activity was measured following the manufacturer’s instruction and is in the Materials and Methods under “**Intracellular ATP Content, Mitochondrial Complex I Activity, and Metabolic Phenotype**”.

13. Line 226 should be revised to include “since the NADH levels increased more than the NAD levels decreased,(a significantly lower ratio of NAD/NADH.

Response:

As the reviewer suggested, this statement was revised as “Since the NADH levels increased more than the NAD levels decreased, a significantly lower ratio of NAD⁺/NADH for cells at P12 was observed compared to cells at P4 (Figure 4B).”

14. Line 269 change increase MMP to hyperpolarize or repolarize MMP. Also need to normalize to mitochondrial mass.

Response:

As the reviewer suggested, this statement has been revised as “Consistent with this, P12 hMSCs with NAM treatment was found to exhibit increased basal autophagy level (**Figure 6G**), as well as the reduction of mitochondrial mass (**Figure 6H**), re-polarized mitochondrial membrane (increased TMRM staining in **Figure 6I**) and increased ETC-I activity (**Figure 6J**).” in the manuscript. To keep results constant, we did not normalize MMP measurement to mitochondrial mass. We already observed the decrease of mito-mass as well as the increase of MMP after NAM treatment. This improvement will be even more announced if MMP was normalized to mass.

15. Discussion line 321-322 states “For the first time, our results demonstrate a close link between gradual loss of basal autophagy and mitophagy, a hallmark of cellular homeostasis 44, 45, and the replicative senescence in hMSCs.” A summary of this finding belongs in the abstract as well.

Response:

As the reviewer suggested, the summary of our key findings now has been added to the abstract.

16. Line 357 states “eventually lose the ability to maintain NAD⁺/NADH redox balance.” If lost, they would die. Instead, they shift to a more reducing, less oxidized redox balance.

Response:

As the reviewer suggested, this statement now has been revised as “However, our results show that senescent hMSCs with damaged mitochondria exhibit low ETC-I activity and the compromised ability to maintain NAD⁺/NADH redox balance. This observation indicates that hMSCs with senescence may shift to the more reducing (or less oxidized) NAD⁺/NADH cycle.”

17. Fig 5A part of legend is cropped for passage number.

Response:

We apologize for this cropped passage number. As the reviewer pointed out, Figure 5A now has been revised. The passage numbers of different groups can be clearly seen. In addition, we further polished Figure 5 to better readability.

18. Lines 369-374 discuss the decline in NAD levels despite a rise in NAMPT. Could it be that the rise in NAMPT is insufficient to metabolize the larger rises in Sirt1 and Sirt3?

Response:

Thanks for raising this possibility. The NAMPT result in our study is a surprise as we are confident on the measurement. The reason we interpreted this as the insufficient substrate is based on the nature relations between NAD⁺, Sirtuins, and NAMPT. The Sirtuins are NAD⁺ dependent and metabolize NAD⁺ to NAM and 2'-O-acetyl-ADPR (Canto et al. 2012). Less NAD⁺ leads to less Sirtuins and thus potentially decreases the production of NAM. Besides, our short-term treatment with NAM did show increase of NAD⁺ as well as Sirt-1/-3. Together, we generated the explanation. As the reviewer suggested, we also include the explanation “Alternatively, the rise in NAMPT may be insufficient to metabolize the larger rises in Sirt1 and Sirt3.” Indeed, more specific studies have to be done to further understand this case in hMSCs or other adult stem cells.

Reference:

Canto et al. Targeting Sirtuin 1 to Improve Metabolism: All You Need Is NAD⁺? Pharmacol Rev. 2012, 64(1):166-187.

19. The methods should state the culture gases, probably 5% CO₂ and 20% oxygen (hyperoxia). The discussion should mention that this alone could account for much of the physiologic

response. Rather than stating the advantages of hypoxia in the discussion, they might be more accurately termed normoxia.

Response:

As the reviewer suggested, the gas composition was added to the Materials and Methods. As mentioned, the culture conditions in this study is the most-commonly used strategies for in vitro culture expansion of hMSCs. The replicative senescence indeed came from the physiological response of hMSCs to this culture environment (both over-enriched nutrients and oxygen level). Relatively, the gas composition (20% O₂) in this study can be termed as normoxia in hypoxic studies as the physiological O₂ level would be lower than 20% to create hypoxic condition. Studies from other research groups as well as our group have demonstrated that hypoxic culture condition improves hMSC proliferation and its therapeutic potentials due to reconfiguration of metabolism. However, it is challenging to maintain robust hypoxia in large-scale production of hMSCs in biomanufacturing. Thus, though there are advantages of hypoxia culture, our NAD⁺ repletion still provides an easy, robust alternative for hMSC *in vitro* expansion.

20. Some comment in the discussion may be needed about why OCR goes up but ETC-I levels go down.

Response:

We appreciated the comments. In general, an increase in basal OCR may come from an increase in ATP turnover, an increase in proton leak, and an increase in non-mitochondrial oxygen consumption such as ROS formation. Indeed, OCR increase should be accompanied by increased ETC-I activity, but it also may come from significant increase of cellular ROS as shown in our study. A statement and references of this possibility has been added to the discussion in the revised manuscript.

21. Supplementary figure S15 needs the colors explained in a legend or figure caption.

Response:

As suggested by the reviewer, the color of each peak in Figure S15, now as **Supplementary Figure S16**, was explained in the revised SI figure legends.

Reviewers' Comments:

Reviewer #1:

Remarks to the Author:

After revision, this manuscript has been improved. However there are still certain minor points remaining to be addressed:

1. P4, last paragraph. The authors wrote "this study tested the hypothesis that culture-induced senescence and metabolic alterations in hMSCs and is correlated with the loss of NAD⁺ homeostasis, which results in the reduction of Sirt-1 signaling". Their hypothesis looks poorly developed and to some degrees confusing. It has also grammar error. The authors need to rephrase it.
2. P5, line 139-142, the sentence "For immunomodulatory functions of hMSCs, mRNA expression of NF- κ B and COX2 were found to be increased in P12 cells compared to P5 cells while the immunosuppressive potential decreased after interferon- γ priming (Supplementary Figure S1)" has grammar error and needs to be corrected.
3. P6, top paragraph. The authors observed decreased secretion of potent anti-inflammatory cytokines HGF 6 143 and IL-10 but increased expression of the pro-inflammatory and anti-angiogenic CXCL10 secretion 144 in late passage of hMSCs. However, there was no difference in terms of IL-6, TNF- α , and IL-1 β secretion between P5 and P12 cells (Supplementary Figure S2). The term "secretion" could be misleading here. What reported in this paper is actually the measurement of cytokine levels in the cultures. They had no evidence as to whether the cytokines were secreted or accumulated due to reduction in degradation. Did the hMSC respond to interferon- γ stimulation by alternation of NAD⁺ homeostasis and Sirt-1 signaling?
4. P6, line 171-1, the authors reported that glycolytic ATP production was found to be significantly reduced in hMSCs at P12 though slight increase of total ATP was also observed (Figure 3A). The result looks contradictory. They may need to explain it in a balanced manner.

Reviewer #2:

Remarks to the Author:

The authors have responded well to the reviewers' critiques with this major revision. The discussion is especially clear now. Only a few minor issues and a number of grammar or syntax issues that need to be addressed.

1. The title seems like an appropriate summary of the findings. But at the end of the abstract, "identify NAD⁺ as a metabolic indicator" seems less impactful. Please consider changing "indicator" to "regulator"
2. Figure 3H, open and closed symbols need to be defined in the legend.
3. Fig. 6F, methods, legend and caption are unclear. How was glycolytic ATP determined? What is ratio? Results on p. 10 could use some explanation more than "NAM increased glycolytic ATP production in P12 cells (Figure 6F)."
4. P. 10, end of first paragraph. "These improvements lead to the partial recovery from replicative senescence in hMSCs at high passage, as well as a shift towards glycolytic energy phenotype." Not clear how data support the conclusion that there is a shift toward a glycolytic phenotype.
5. Numerous grammar problems and awkward syntax:
 - a. Three in the abstract alone should read: "play a regulatory role in cellular senescence of hMSCs. Treatment with the NAD⁺ precursor nicotinamide...", "partially recovered resulted in partial recovery of mitochondrial fitness"
 - b. Top of p. 4, in vitro culture is redundant. Choose one.
 - c. P. 7, line 7, meaning is unclear: "as the carbon of glucose has higher flux in TCA cycle"
 - d. dissimilarity in the proteome
 - e. p.7 toward bottom. (How many?) proteins showing more than 10-fold change were plotted in

volcano plots, which include those associated with NAD metabolism (e.g., HADHA, NNMT, HSD17B4, OGDH and DLD) (Supplementary Figure S4C).

f. p. 10, near top is awkward, "The cell cycle arrest was eased to a certain level after adding NAM as the increase of cell population in S phase was shown.

g. Bottom p. 10. "NAD⁺/NADH ratio remained relatively constant was relatively maintained at constant.

h. P. 11. For multipotency, senescent hMSCs tends to have lower ability for differentiation but conflict observations have been published. Something like this would be better. "Some reports describe a lower ability for differentiation and decreased multipotency of senescent hMSCs."

Response to Reviewers' Comments

Again, the authors appreciate the thoughtful and constructive comments by the reviewers. We have thoroughly revised the manuscript and added detailed information/discussions to address the comments from the reviewers. The detailed responses to the comments are summarized in the following section and the revisions in the manuscript and supplementary information are highlighted in red color. The reviewers' comments are reproduced in *italic font*, and our responses are in regular font.

Reviewer #1 (Remarks to the Author): MS# COMMSBIO-20-0427-T Li et al. NAD⁺/NADH redox

After revision, this manuscript has been improved. However there are still certain minor points remaining to be addressed:

1. P4, last paragraph. The authors wrote "this study tested the hypothesis that culture-induced senescence and metabolic alterations in hMSCs and is correlated with the loss of NAD⁺ homeostasis, which results in the reduction of Sirt-1 signaling". Their hypothesis looks poorly developed and to some degrees confusing. It has also grammar error. The authors need to rephrase it.

Response:

We thank the reviewer for the suggestions. We have revised the hypothesis and make it more clear towards our experimental design and background knowledge. This part now reads as "this study tested the hypothesis that in vitro culture expansion induces replicative senescence and metabolic alterations in hMSCs which correlate with the loss of NAD⁺ homeostasis and result in the reduction of Sirt-1 signaling activity."

2. P5, line 139-142, the sentence "For immunomodulatory functions of hMSCs, mRNA expression of NF- κ B and COX2 were found to be increased in P12 cells compared to P5 cells while the immunosuppressive potential decreased after interferon- γ priming (Supplementary Figure S1)" has grammar error and needs to be corrected.

Response:

As the reviewer suggested, this statement has been revised as "For immunomodulation of hMSCs, mRNA expression of genes (NF- κ B and COX2) involved in chronic inflammation was found to be upregulated in P12 hMSCs compared to P5 cells, while the immunosuppressive ability via IDO pathways in P12 hMSCs decreased (2.76-fold vs. 4.41-fold) under interferon- γ priming (Supplementary Figure S1)."

3. P6, top paragraph. The authors observed decreased secretion of potent anti-inflammatory cytokines HGF 6 143 and IL-10 but increased expression of the pro-inflammatory and anti-angiogenic CXCL10 secretion 144 in late passage of hMSCs. However, there was no difference in terms of IL-6, TNF- α , and IL-1 β secretion between P5 and P12 cells (Supplementary Figure S2). The term "secretion" could be misleading here. What reported in this paper is actually the measurement of cytokine levels in the cultures. They had no evidence as to whether the cytokines were secreted or accumulated due to reduction in degradation. Did the hMSC respond to interferon- γ stimulation by alternation of NAD⁺ homeostasis and Sirt-1 signaling?

Response:

We agree with the reviewer and this is a valid point for discussion. First, we revised the statement for the cytokine level during in vitro culture. Second, we have tested the IFN- γ priming after treatment of NAM, however, there is no significant difference for IDO activity under IFN- γ priming between P12 and P12+NAM groups, thus the results are not incorporated in this manuscript. Last, we suspect that the culture-induced cellular senescence of hMSCs is similar to chronic inflammation in stem cells as the pro-inflammatory cytokines accumulated and the basal IDO activity are relatively higher in late passage of hMSCs.

4. P6, line 171-1, the authors reported that glycolytic ATP production was found to be significantly reduced in hMSCs at P12 though slight increase of total ATP was also observed (Figure 3A). The result looks contradictory. They may need to explain it in a balanced manner.

Response:

Actually, the decrease of glycolytic ATP not necessarily leads to the decrease of total ATP. Cellular ATP can be generated via multiple metabolic pathways such as pentose phosphate pathway (PPP), fatty acid β -oxidation and glutaminolysis. It is possible that hMSCs at late passage employ other metabolic pathways to generate energy. In fact, our proteomics analysis indicates the potential activation of fatty acid β -oxidation in P12 hMSCs compared to P4 cells. Corresponding statements have been added to the discussion in the revised manuscript.

Reviewer #2 (Remarks to the Author):

The authors have responded well to the reviewers' critiques with this major revision. The discussion is especially clear now. Only a few minor issues and a number of grammar or syntax issues that need to be addressed.

1. The title seems like an appropriate summary of the findings. But at the end of the abstract, "identify NAD⁺ as a metabolic indicator" seems less impactful. Please consider changing "indicator" to "regulator"

Response:

We thank the reviewer for the suggestions. The abstract has been revised as suggested as well as requested by the editorial office.

2. Figure 3H, open and closed symbols need to be defined in the legend.

Response:

We thank the reviewer for the suggestions. The figure legend has been revised in Figure 3.

3. Fig. 6F, methods, legend and caption are unclear. How was glycolytic ATP determined? What is ratio? Results on p. 10 could use some explanation more than "NAM increased glycolytic ATP production in P12 cells (Figure 6F)."

Response:

We agree with the reviewer's comments. The detailed methods on how to determine glycolytic ATP ratio has been added to Methods section in the revised manuscript. The ratio represents the

percentage of glycolytic ATP in the total ATP production. Additional interpretation of Figure 6H has also been added in the revised manuscript.

4. P. 10, end of first paragraph. *“These improvements lead to the partial recovery from replicative senescence in hMSCs at high passage, as well as a shift towards glycolytic energy phenotype.”* Not clear how data support the conclusion that there is a shift toward a glycolytic phenotype.

Response:

We agree with that the evidence for the shift toward glycolytic phenotype is limited (only Glycolytic ATP). Thus, we revised the summary for this figure to “These improvements lead to the partial recovery from replicative senescence in hMSCs at high passage, as well as the improved mitochondrial fitness that facilitates central energy metabolism.”

5. Numerous grammar problems and awkward syntax:

a. Three in the abstract alone should read: *“play a regulatory role in cellular senescence of hMSCs. Treatment with the NAD⁺ precursor nicotinamide...”, “partially recovered resulted in partial recovery of mitochondrial fitness”*

Response:

This statement has been revised as suggested, as well as requested by editorial office: “We show that alterations of cellular nicotinamide adenine dinucleotide (NAD⁺ /NADH) redox balance and activity of the Sirtuin (Sirt) family enzymes regulate cellular senescence of hMSCs. Treatment with NAD⁺ precursor nicotinamide increases the intracellular NAD⁺ level and re-balances the NAD⁺ /NADH ratio, with enhanced Sirt-1 activity in hMSCs at high passage, partially restores mitochondrial fitness and rejuvenates senescent hMSCs.”

b. Top of p. 4, *in vitro culture is redundant. Choose one.*

Response:

As the reviewer suggested, we revised the expression of “in vitro culture expansion” as “in vitro expansion”.

c. P. 7, line 7, meaning is unclear: *“as the carbon of glucose has higher flux in TCA cycle”*

Response:

As the reviewer suggested, this statement has been revised as “suggesting an increased coupling of glycolysis and TCA cycle metabolism as the labeled carbon of glucose has higher enrichment in metabolites from TCA cycle.”

d. *dissimilarity in the proteome*

Response:

As the reviewer pointed out, this statement has been revised as “...to illustrate the global difference in proteome of hMSCs at early passage and late passage with replicative senescence.”

e. p.7 toward bottom. (How many?) proteins showing more than 10-fold change were plotted in volcano plots, which include those associated with NAD metabolism (e.g., HADHA, NNMT, HSD17B4, OGDH and DLD) (Supplementary Figure S4C).

Response:

As the reviewer suggested, this part has been revised in the manuscript as “The 587 proteins were plotted in volcano plots and those with more than 10-fold change and those associated with NAD metabolism (e.g., HADHA, NNMT, HSD17B4, OGDH and DLD) were marked as yellow”.

f. p. 10, near top is awkward, “The cell cycle arrest was eased to a certain level after adding NAM as the increase of cell population in S phase was shown.

Response:

As the reviewer suggested, this statement has been revised as “After adding NAM, the increase of cell population in the S phase of cell cycle was observed.”

g. Bottom p. 10. “NAD⁺/NADH ratio remained relatively constant was relatively maintained at constant.

Response:

As the reviewer suggested, this statement has been revised as “More interestingly, intracellular NAD⁺ level and NAD⁺/NADH ratio was relatively stable during culture expansion up to 15 passages.”

h. P. 11. For multipotency, senescent hMSCs tends to have lower ability for differentiation but conflict observations have been published. Something like this would be better. “Some reports describe a lower ability for differentiation and decreased multipotency of senescent hMSCs.”

Response:

As the reviewer suggested, this statement has been revised as “Some reports describe a lower ability for differentiation and decreased multipotency of senescent hMSCs. For example, some studies.....”